# Developmental NMDA receptor dysregulation in the infantile neuronal ceroid lipofuscinosis mouse model

Kevin P Koster[1], Walter Francesconi[1], Fulvia Berton[1], Sami Alahmadi[1], Roshan Srinivas[1], Akira Yoshii[2,3]*

[1]Department of Anatomy and Cell Biology, University of Illinois at Chicago, Chicago, United States; [2]Department of Pediatrics, University of Illinois at Chicago, Chicago, United States; [3]Department of Neurology, University of Illinois at Chicago, Chicago, United States

**Abstract** Protein palmitoylation and depalmitoylation alter protein function. This post-translational modification is critical for synaptic transmission and plasticity. Mutation of the depalmitoylating enzyme palmitoyl-protein thioesterase 1 (PPT1) causes infantile neuronal ceroid lipofuscinosis (CLN1), a pediatric neurodegenerative disease. However, the role of protein depalmitoylation in synaptic maturation is unknown. Therefore, we studied synapse development in $Ppt1^{-/-}$ mouse visual cortex. We demonstrate that the developmental N-methyl-D-aspartate receptor (NMDAR) subunit switch from GluN2B to GluN2A is stagnated in $Ppt1^{-/-}$ mice. Correspondingly, $Ppt1^{-/-}$ neurons exhibit immature evoked NMDAR currents and dendritic spine morphology in vivo. Further, dissociated $Ppt1^{-/-}$ cultured neurons show extrasynaptic, diffuse calcium influxes and enhanced vulnerability to NMDA-induced excitotoxicity, reflecting the predominance of GluN2B-containing receptors. Remarkably, $Ppt1^{-/-}$ neurons demonstrate hyperpalmitoylation of GluN2B as well as Fyn kinase, which regulates surface retention of GluN2B. Thus, PPT1 plays a critical role in postsynapse maturation by facilitating the GluN2 subunit switch and proteostasis of palmitoylated proteins.

DOI: https://doi.org/10.7554/eLife.40316.001

*For correspondence:
ayoshii@uic.edu

Competing interests: The authors declare that no competing interests exist.

## Introduction

The neuronal ceroid lipofuscinoses (NCLs) are a class of individually rare, primarily autosomal reces-sive, neurodegenerative diseases occurring in an estimated 2 to 4 of 100,000 live births (*Nita et al., 2016*). Collectively, NCLs represent the most prevalent class of hereditary pediatric neurodegenera-tive disease (*Haltia, 2006*). The NCLs are characterized by progressive neurodegeneration, blind-ness, cognitive and motor deterioration, seizures, and premature death. The cardinal feature of all NCLs is the intracellular accumulation of proteolipid material, termed lipofuscin (*Jalanko and Braulke, 2009*; *Nita et al., 2016*). While lipofuscin accumulates in all cells of affected individuals, it deposits most robustly in neurons. This accumulation is concurrent with rapid and progressive neuro-degeneration, particularly of thalamic and primary sensory cortical areas (*Bible et al., 2004*; *Kielar et al., 2007*). The NCLs are categorized into *CLN1-14* based on the age of onset and the causative gene mutated. The products of *CLN* genes are lysosomal and endosomal proteins, there-fore NCLs are also classified as lysosomal storage disorders (LSDs) (*Bennett and Hofmann, 1999*; *Jalanko and Braulke, 2009*). The infantile form of disease, CLN1, presents as early as 6 months of age with progressive psychomotor deterioration, seizure, and death at approximately 5 years of age (*Haltia, 2006*; *Jalanko and Braulke, 2009*; *Nita et al., 2016*). CLN1 disease is caused by mutations in the gene *CLN1*, which encodes the enzyme palmitoyl-protein thioesterase 1 (PPT1)(*Camp and*

*Hofmann, 1993*; *Camp et al., 1994*; *Vesa et al., 1995*; *Jalanko and Braulke, 2009*). PPT1 is a depalmitoylating enzyme responsible for the removal of palmitic acid from modified proteins (*Camp and Hofmann, 1993*; *Lu and Hofmann, 2006*).

Protein palmitoylation, the addition of a 16-carbon fatty acid (palmitic acid) to cysteine residues, is a crucial regulator of protein trafficking and function, particularly in neurons (*Hayashi et al., 2005*; *Hayashi et al., 2009*; *Fukata et al., 2006*; *Kang et al., 2008*; *Fukata and Fukata, 2010*; *Han et al., 2015*). This post-translational modification is mediated by palmitoyl acyltransferases (PATs) of the DHHC enzyme family (*Fukata et al., 2006*; *Fukata and Fukata, 2010*). In contrast to other types of protein acylation, palmitoylation occurs via a reversible thioester bond (s-palmitoylation), permitting dynamic control over target protein interactions and function. Further, palmitoylated proteins require depalmitoylation prior to lysosomal degradation (*Lu et al., 1996*; *Lu and Hofmann, 2006*). Consequently, protein palmitoylation and depalmitoylation contribute significantly to mechanisms underlying synaptic plasticity and endosomal-lysosomal trafficking of proteins (*Hayashi et al., 2005*; *Hayashi et al., 2009*; *Kang et al., 2008*; *Lin et al., 2009*; *Noritake et al., 2009*; *Fukata and Fukata, 2010*; *Mattison et al., 2012*; *Thomas et al., 2012*; *Thomas et al., 2013*; *Fukata et al., 2013*; *Han et al., 2015*). Indeed, PPT1 is a lysosomal depalmitoylating enzyme that localizes to the axonal and synaptic compartments (*Verkruyse and Hofmann, 1996*; *Ahtiainen et al., 2003*; *Kim et al., 2008*). The synaptic association of PPT1 and prominence of palmitoylated synaptic proteins suggests that PPT1 influences synaptic functions through, at least, protein turnover. Many synaptic proteins undergo palmitoylation, including, but not limited to postsynaptic density protein 95 (PSD-95), all GluA subunits of AMPARs, and the GluN2A/2B subunits of NMDARs (*Kang et al., 2008*). However, the role of depalmitoylation in regulating synaptic protein function remains less clear.

N-methyl-D-aspartate receptors (NMDARs) are voltage-dependent, glutamate-gated ion channels consisting of two obligatory GluN1 subunits and two GluN2 subunits that undergo a developmental change (*Cull-Candy et al., 2001*; *van Zundert et al., 2004*; *Lau and Zukin, 2007*; *Paoletti et al., 2013*). NMDARs play a crucial role in synaptic transmission, postsynaptic signal integration, synaptic plasticity, and have been implicated in various neurodevelopmental and psychiatric disorders (*Lau and Zukin, 2007*; *Lakhan et al., 2013*; *Paoletti et al., 2013*; *Yamamoto et al., 2015*). NMDAR subunit composition, receptor localization, and downstream signaling mechanism undergo developmental regulation (*Watanabe et al., 1992*; *Monyer et al., 1994*; *Sheng et al., 1994*; *Li et al., 1998*; *Stocca and Vicini, 1998*; *Tovar and Westbrook, 1999*; *Losi et al., 2003*; *van Zundert et al., 2004*; *Paoletti et al., 2013*; *Wyllie et al., 2013*). Specifically, GluN2B-containing NMDARs are expressed neonatally and display prolonged decay kinetics, which allows comparatively increased calcium influx thought to facilitate forms of synaptic plasticity critical for neurodevelopment (*Sobczyk et al., 2005*; *Zhao et al., 2005*; *Zhao et al., 2013*; *Zhang et al., 2008*; *Evans et al., 2012*; *Shipton and Paulsen, 2014*). These GluN2B-containing receptors are supplanted at the synapse by diheteromeric GluN1/GluN2A NMDARs or triheteromeric (GluN1/GluN2A/GluN2B) receptors in response to experience-dependent neuronal activity (*Quinlan et al., 1999b*; *Quinlan et al., 1999a*; *Tovar and Westbrook, 1999*; *Philpot et al., 2001*; *Liu et al., 2004*; *Paoletti et al., 2013*; *Tovar et al., 2013*). This developmental switch of GluN2B- to GluN2A-containing NMDARs during brain maturation is mediated by the postsynaptic scaffolding receptors, SAP102 and PSD-95, respectively; SAP102-GluN2B-NMDAR complexes are replaced by PSD-95-GluN2A-NMDAR complexes in response to developmental, experience-dependent activity (*Sans et al., 2000*; *van Zundert et al., 2004*; *Elias et al., 2008*). While PSD-95, GluN2B, and GluN2A all undergo palmitoylation, how depalmitoylation regulates the turnover of these proteins, let alone during the GluN2B to GluN2A subunit switch, is unclear.

In the current study, we investigated the cellular and synaptic effects of PPT1-deficiency using the $Ppt1^{-/-}$ mouse model of CLN1 disease. We focused on the visual system in $Ppt1^{-/-}$ animals for two reasons. First, cortical blindness is a characteristic feature of CLN1 disease. Second, the rodent visual system is a well-studied model of cortical development and synaptic plasticity/maturation and it therefore serves as an optimal experimental model to examine the role of PPT1-mediated depalmitoylation during development. We found that lipofuscin accumulated in the $Ppt1^{-/-}$ visual cortex shortly after eye-opening at postnatal day (P) 14, a timing earlier than previously documented (*Gupta et al., 2001*). Using biochemistry and electrophysiology, we found impeded developmental NMDAR subunit switch from GluN2B to GluN2A in $Ppt1^{-/-}$ mice compared to wild-type (WT). This NMDAR disruption is associated with disrupted dendritic spine morphology *in vivo*. To gain further mechanistic insight into neurodegeneration in CLN1, we used cultured cortical neurons and found

that *Ppt1$^{-/-}$* cells recapitulate the disrupted dendritic spine phenotype and GluN2B to GluN2A switch, leading to excessive extrasynaptic calcium transients and enhanced vulnerability to NMDA-mediated excitotoxicity. We directly examined protein palmitoylation state and found hyperpalmitoylation of GluN2B as well as Fyn kinase, which facilitates GluN2B surface retention, in *Ppt1$^{-/-}$* neurons. Finally, we demonstrate that chronic treatment of *Ppt1$^{-/-}$* neurons with palmitoylation inhibitors normalized GluN2B and Fyn kinase hyperpalmitoylation and rescued the enhanced susceptibility to excitotoxicity. Our results indicate that PPT1 plays a critical role in the developmental GluN2B to GluN2A subunit switch and synaptic maturation. Further, our results indicate that these dysregulated mechanisms contribute to CLN1 pathophysiology and may be shared features of common adult-onset neurodegenerative diseases.

## Results

To understand synaptic dysregulation in CLN1 disease, we utilized the visual cortex of *Ppt1$^{-/-}$* animals as a model system. The rodent visual cortex undergoes timed, experience-dependent plasticity, which has been well-characterized at the systemic, cellular, and molecular levels (*Bear et al., 1990*; *Gordon and Stryker, 1996*; *Hensch et al., 1998*; *Quinlan et al., 1999a*; *Fagiolini and Hensch, 2000*; *Mataga et al., 2001*; *Mataga et al., 2004*; *Philpot et al., 2001*; *Desai et al., 2002*; *Yoshii et al., 2003*; *Hensch, 2005*; *Cooke and Bear, 2010*). We examined WT and *Ppt1$^{-/-}$* littermates at the following ages: P11, P14, P28, P33, P42, P60, P78, P120, which correspond to particular developmental events in visual cortex. In mice, P11 and P14 are prior to and just after eye opening (EO), respectively. Further, the critical period in the visual cortex peaks at P28 and closes from P33 to P42. We chose postnatal day 60, P78, and P120 were selected as adult time points. We determined whether experience-dependent synaptic maturation is altered during the progression of CLN1 pathology.

### Lipofuscin deposits immediately following eye opening in visual cortex of *Ppt1$^{-/-}$* mice

Although it remains controversial whether lipofuscin is toxic to neurons or an adaptive, neuroprotective mechanism, its accumulation correlates with disease progression. Therefore, we examined lipofuscin deposition in the visual cortex as a marker of pathology onset and progression. Lipofuscin aggregates are readily detectable as autofluorescent lipopigments (ALs) without staining under a confocal microscope. To examine the temporal and spatial accumulation of ALs in *Ppt1$^{-/-}$* mice, we performed quantitative histology on the visual cortex (area V1) of WT and *Ppt1$^{-/-}$* mice during early development. Visual cortical sections were imaged at the above-mentioned developmental time points, and ALs were quantified in a laminar-specific manner. We found that ALs are detectable first at P14 in *Ppt1$^{-/-}$* visual cortex, earlier than previously reported at 3 or 6 months (*Figure 1A–C*) (*Gupta et al., 2001*; *Blom et al., 2013*). Further, ALs accumulated rapidly through the critical period (*Berardi et al., 2000*; *Hensch, 2005*; *Maffei and Turrigiano, 2008*) and plateaued by adulthood (P60) (*Figure 1A–C*). This result suggests that neuronal AL load is saturable, and that this saturation occurs early on in disease, as *Ppt1$^{-/-}$* animals do not perish until around 10 months of age.

Whether lipofuscin accumulation is directly neurotoxic or not, profiling the temporospatial and sub-regional pattern of AL deposition will be valuable for assessing therapeutic interventions in future studies. The pattern of deposition revealed herein suggests a correlation between systemic neuronal activation and AL accumulation, as we found that AL deposition started immediately following EO, the onset of patterned visual activity, and accumulated rapidly during development (*Figure 1A,C*, *Supplementary file 1*). These findings suggest that neuronal activity or experience-dependent plasticity may be linked to lipofuscin deposition.

### NMDAR subunit composition is biased toward immaturity in *Ppt1$^{-/-}$* visual cortex

To examine the role of PPT1 in excitatory synapse function, we focused on the NMDAR subunits, GluN2B and GluN2A, which are both palmitoylated (*Hayashi et al., 2009*). The developmental GluN2B to GluN2A subunit change (*Paoletti et al., 2013*) is critical for NMDAR function and maturation, which facilitates refinement of neural circuits and a higher tolerance to glutamate-mediated excitotoxicity (*Hardingham and Bading, 2002*; *Hardingham and Bading, 2010*; *Hardingham et al.,*

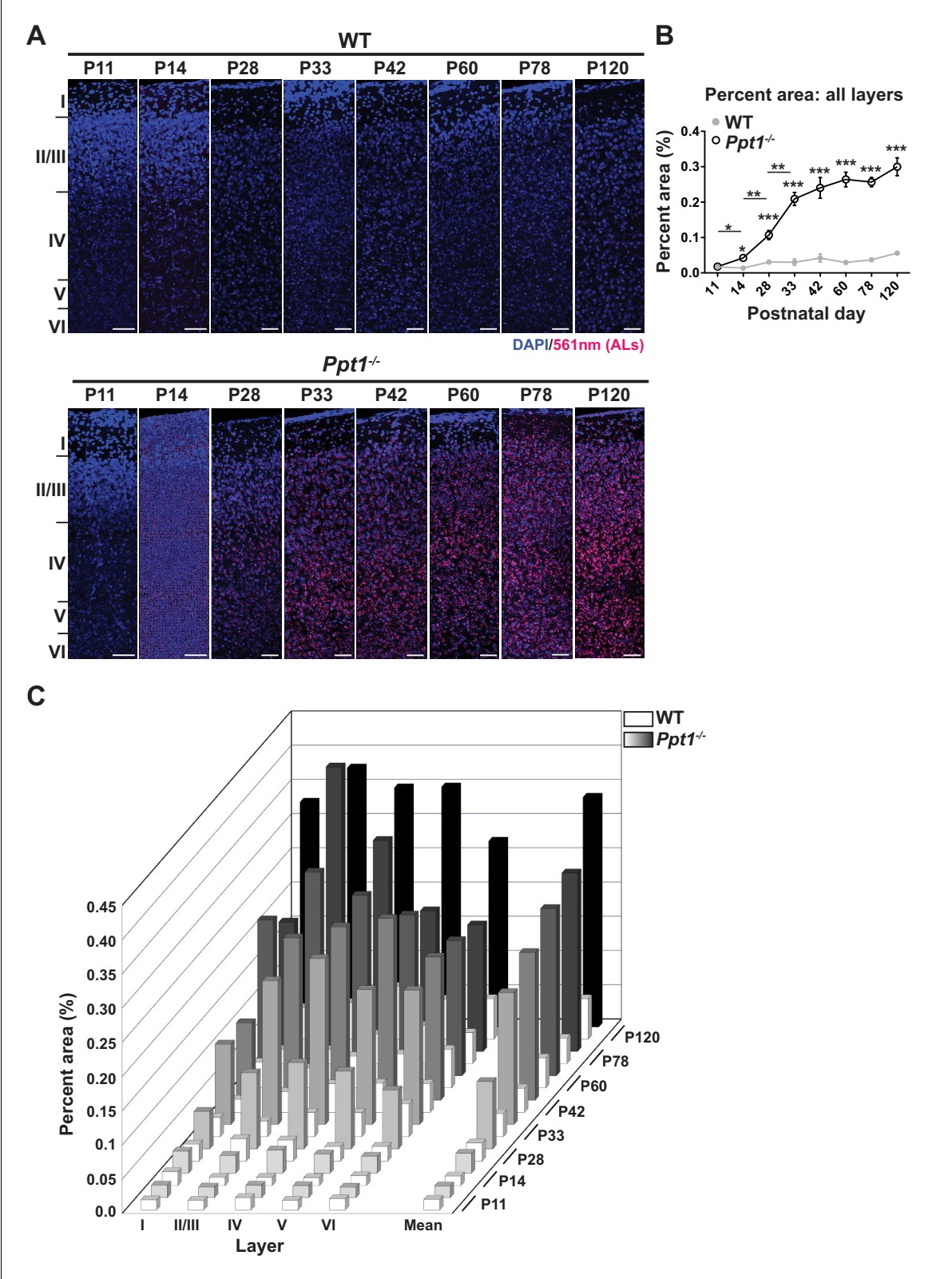

**Figure 1.** ALs deposit immediately following eye opening in visual cortex of *Ppt1*[-/-] mice. (A) Representative composite confocal images through area V1 of visual cortex in WT (top) and *Ppt1*[-/-] mice (bottom) during development and into adulthood. DAPI nuclear stain (blue, 405 nm excitation) and AL signals (red, 561 nm excitation) are visualized. Cortical layers are marked (left). Scale = 50 μm. Note that scale bars for P11 and P14 images are enlarged to account for reduced cortical thickness at these ages. (B) Quantification of the mean percent area occupied by ALs through all cortical layers (see

*Figure 1 continued on next page*

*Figure 1 continued*

Materials and methods). *Ppt1⁻/⁻* and WT were compared (n = 4–6 animals/group) at each age using t-test and the significance was indicated as follows: *p<0.05, **p<0.01, and ***p<0.001. Differences between two consecutive ages (e.g. *Ppt1⁻/⁻* P14 vs. (P11) is denoted: *p<0.05 and **p<0.01 where indicated. Error bars represent s.e.m. (C) Cortical layer-specific quantification of area occupied by ALs separated by each cortical layer (x-axis) and age (z-axis). Averaged values, s.e.m., and n for each condition are represented in **Supplementary file 1**.

DOI: https://doi.org/10.7554/eLife.40316.002

*2002*). Furthermore, previous work shows evidence for NMDA-induced excitotoxicity in the pathogenesis of CLN1 (*Finn et al., 2012*). We biochemically analyzed WT and *Ppt1⁻/⁻* visual cortices from P11 to P60 and measured levels of GluN2B and GluN2A subunits in synaptosomes and whole lysates of WT and *Ppt1⁻/⁻* visual cortices. Although GluN2B levels were comparable between WT and *Ppt1⁻/⁻* at all ages, GluN2A levels in synaptosomes were significantly lower in *Ppt1⁻/⁻* than WT (*Figure 2A*). This decrease was present at time points during, and just following, the critical period in visual cortical development (P33, P42, and P60). When analyzed as a ratio of GluN2A/GluN2B, a robust and persistent decrease is observed in *Ppt1⁻/⁻* visual cortex (*Figure 2B*). GluN1 levels were unchanged between WT and *Ppt1⁻/⁻* in synaptosomes (*Figure 2C*), indicating the selective obstruction of GluN2A incorporation into NMDARs.

The developmental shift from GluN2B-containing NMDARs to synaptic GluN2A-containing NMDARs is mediated by the postsynaptic scaffolding proteins, SAP102 and PSD-95 (*Townsend et al., 2003*; *van Zundert et al., 2004*; *Elias et al., 2008*). SAP102 preferentially interacts with GluN2B-containing NMDARs, which are enriched neonatally (*Sans et al., 2000*; *van Zundert et al., 2004*; *Zheng et al., 2010*; *Chen et al., 2011*). In contrast, PSD-95 has greater affinity to GluN2A-containing NMDARs, particularly in the mature brain (*Sans et al., 2000*; *van Zundert et al., 2004*; *Dongen, 2009*; *Yan et al., 2014*). Thus, we examined the expression of these scaffolding proteins in WT and *Ppt1⁻/⁻* visual cortex. While SAP102 levels remained unchanged, PSD-95 levels reduced at P33-P60, the same developmental time points where GluN2A expression also decreased (*Figure 2D*). Together, these results suggest reduced incorporation and scaffolding of GluN2A-containing NMDARs in *Ppt1⁻/⁻* synapses, indicating immature or dysfunctional synaptic composition.

Next, we measured PPT1 protein level across the same time points in WT animals to examine whether the expression profile of PPT1, and presumably its cellular activity, temporally correlated with the observed reductions in mature synaptic components in *Ppt1⁻/⁻* animals. Indeed, PPT1 expression in synaptosomes is low at P11 and P14 and increases with age, reaching peak levels between P33 and P60 (*Figure 2E*). This expression profile correlates with the time course of AL accumulation (*Figure 1B and C*) and fits with the notion that PPT1 activity at the synapse plays a role in neurodevelopmental processes.

To examine whether the reduction in GluN2A is due to selective exclusion from the postsynaptic site or alterations in the total protein amount, we also measured NMDAR subunit levels in whole lysates. These findings closely match our findings in synaptosomes. Namely, GluN2A levels showed reductions in *Ppt1⁻/⁻* lysates beginning at the same time point (P33) (*Figure 2—figure supplement 1A*), while GluN2B levels were stable (*Figure 2—figure supplement 1B*). The GluN2A/2B ratios in *Ppt1⁻/⁻* whole lysates were also lower than those in WT lysates and the reduction was comparable to that observed in synaptosomes (*Figure 2—figure supplement 1C*). GluN1 levels, however, were unaltered between genotypes (*Figure 2—figure supplement 1D*), again indicating a selective reduction in the expression of GluN2A. Interestingly, while PPT1 levels in whole lysates were, similarly to synaptosomes, low at P11 and P14, expression subsequently peaked at P28 and P33 before declining at P60 (*Figure 2—figure supplement 1E*). Together, these results indicate a selective decrease in the total amount of mature synaptic components in *Ppt1⁻/⁻* brains that temporally correlates with the cellular PPT1 expression profile in developing WT neurons. Furthermore, our findings in whole lysates suggest that synaptosomal reductions in GluN2A and PSD-95 may result from altered transcription or translation of these proteins instead of direct depalmitoylation by PPT1.

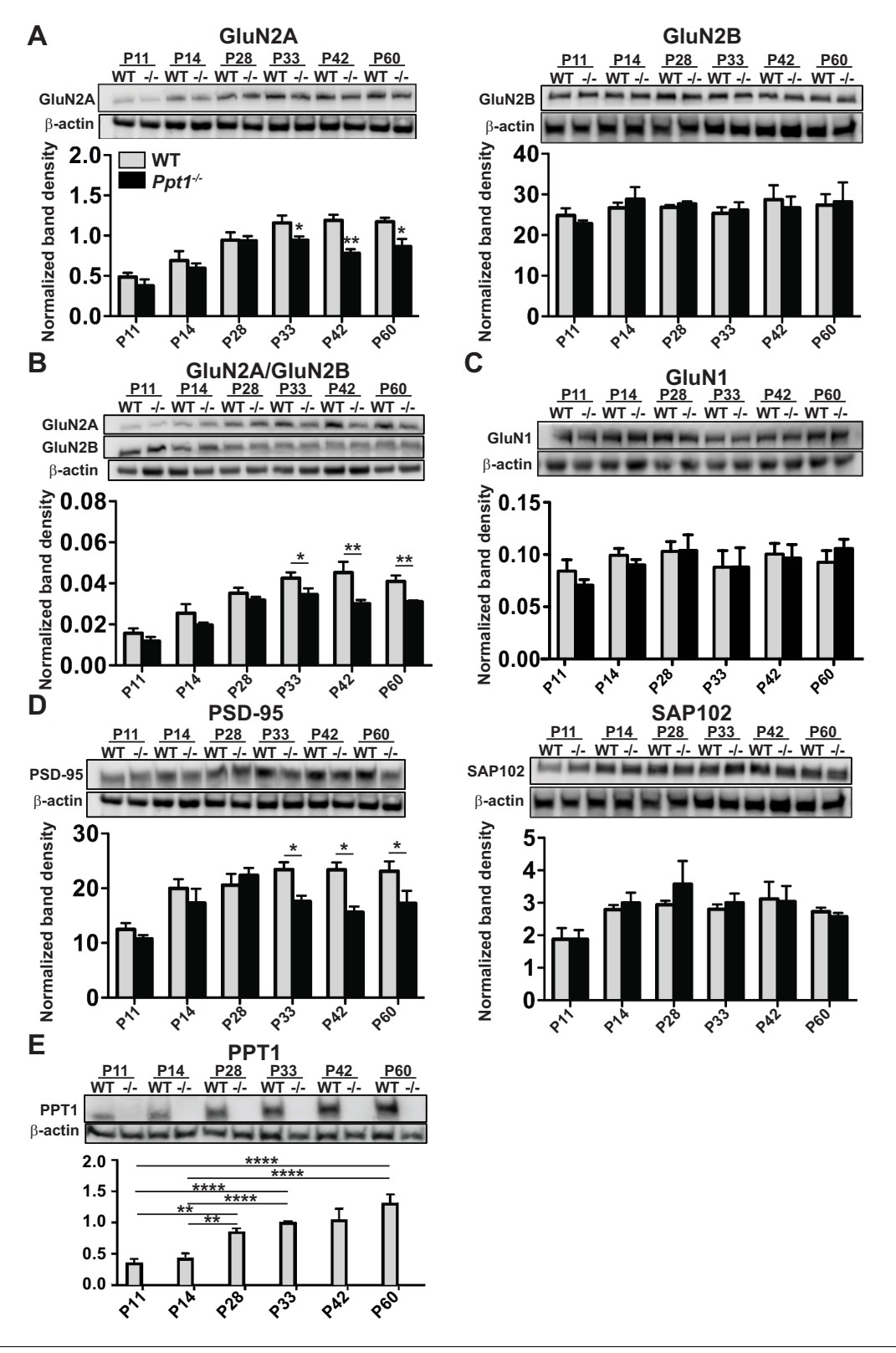

**Figure 2.** NMDAR subunit composition is biased toward immaturity in *Ppt1*-/- visual cortex. (**A**) Representative immunoblots from synaptosomes of GluN2 subunits, GluN2A and GluN2B across age (P11–P60) and genotype as indicated (top) and quantification of band density (bottom) normalized to β-actin loading control within lane. (**B**) Representative immunoblots from synaptosomes of GluN2A and GluN2B (top) and quantification of the ratio of GluN2A/GluN2B band density within animal normalized to β-actin loading control within lane (bottom). (**C**) Representative immunoblot of GluN1 from

*Figure 2 continued on next page*

*Figure 2 continued*

synaptosomes across age and genotype as indicated (top) and quantification of band density (bottom) normalized to β-actin loading control within lane. (D) Representative immunoblots from synaptosomes of the scaffolding molecules PSD-95 and SAP102 across age and genotype as indicated (top) and quantification of band density (bottom) normalized to β-actin loading control within lane. (E) Representative immunoblot from synaptosomes of PPT1 across age and genotype as indicated (top) and protein expression level (bottom) normalized to β-actin. For experiments in *Figure 2A–D*, *Ppt1*$^{-/-}$ and WT were compared (n = 4 independent experiments/animals with two repetitions/group) at each age using t-test and the significance was indicated as follows: *p<0.05, and **p<0.01. In *Figure 2E*, WT expression levels at each age were compared (n = 4 independent experiments/animals with two repetitions/group) by ANOVA followed by Tukey's post-hoc test. Significance between ages is indicated: *p<0.05. Error bars represent s.e.m.

DOI: https://doi.org/10.7554/eLife.40316.003

The following figure supplement is available for figure 2:

**Figure supplement 1.** NMDAR subunit composition is immature in whole lysates from *Ppt1*$^{-/-}$ visual cortex.

DOI: https://doi.org/10.7554/eLife.40316.004

## NMDAR-mediated EPSCs are altered in *Ppt1*$^{-/-}$ visual cortex

Next, we sought to correlate our biochemical findings with electrophysiological changes in NMDAR functionality (*Figure 2*). While human CLN1 patients present with retinal degeneration and the *Ppt1*$^{-/-}$ mouse model of CLN1 phenocopies the human disease, the electroretinogram (ERG) is effectively unaltered at 4 months in the mouse model (*Lei et al., 2006*), allowing for detailed study of the electrophysiological changes in the visual cortex associated with early disease states. We recorded evoked, NMDAR-mediated excitatory postsynaptic currents (EPSCs) in layer II/III cortical neurons in visual cortical slices of WT and *Ppt1*$^{-/-}$ mice at P42. The NMDAR-EPSCs were pharmacologically isolated (see Materials and methods section) and were recorded in whole cell patch mode clamped at +50 mV. As GluN2A- and GluN2B-containing NMDARs exhibit differential receptor kinetics, with GluN2A displaying fast (~50 ms) and GluN2B displaying slow decay kinetics (~300 ms), their relative contribution is reliably interpolated by fitting the EPSC decay phase with a double exponential function (*Stocca and Vicini, 1998*; *Vicini et al., 1998*). From the fitting of absolute amplitude-normalized, WT and *Ppt1*$^{-/-}$ NMDAR-EPSCs (*Figure 3A*), we measured the following parameters: the ratios of the amplitudes (A) of the fast, $A_f/A_f + A_s$, and slow, $A_s/A_f + A_s$ components, and the weighted decay time constants ($\tau_w$). The fast component ($A_f/A_f + A_s$) of NMDAR-ESPC amplitudes decreased in *Ppt1*$^{-/-}$ mice as compared to WT, while the slow component ($A_s/A_f + A_s$) increased (*Figure 3B*). Further, *Ppt1*$^{-/-}$ neurons showed a significant increase in weighted decay time $\tau_w$ as compared to WT (*Figure 3C*). Remarkably, the rise time (time to peak amplitude) of *Ppt1*$^{-/-}$ NMDAR-EPSCs was slightly but significantly longer than WT (*Figure 3D*), suggesting that the response involves the receptors more distant from the presynaptic release site. Indeed, previous studies documented similar observations and postulated that a longer rise time is characteristic of GluN2B-containing NMDARs that are preferentially localized on the extrasynaptic membrane (*Townsend et al., 2003*; *van Zundert et al., 2004*; *Sanz-Clemente et al., 2013*).

Next, we treated cortical slices with Ro 25–6981, a potent and selective inhibitor of GluN2B-containing NMDARs (*Fischer et al., 1997*) and asked if these receptors are overrepresented in *Ppt1*$^{-/-}$ neurons. We recorded NMDAR-EPSCs at baseline and during bath infusion of Ro 25–6981 (30 min, 3µM), then compared the percent inhibition ($\tau_w$ percent of baseline) between WT and *Ppt1*$^{-/-}$ groups. Ro 25–6981 treatment significantly decreased the $\tau_w$ of NMDAR-EPSCs from the baseline in both WT and *Ppt1*$^{-/-}$ cells at P42 (*Figure 3E*). To our surprise, no significant effects were present between the two genotypes after Ro 25–6981 treatment by two-way ANOVA, suggesting that NMDARs are inhibited to the same degree in WT and *Ppt1*$^{-/-}$ cortices.

At first glance, the above result did not fulfill the anticipation that NMDA-EPSCs in *Ppt1*$^{-/-}$ neurons would respond to Ro 25–6981 treatment to a greater degree than in WT, due to an overrepresentation of GluN2B-containing receptors. However, cortical neurons predominantly express NMDARs consisting of two GluN1, one GluN2A and one GluN2B subunits (*Sheng et al., 1994*; *Luo et al., 1997*; *Tovar and Westbrook, 1999*). These triheteromeric NMDARs display prolonged decay kinetics compared to GluN2A-diheteromeric NMDARs, while being largely insensitive to GluN2B-specific antagonists (*Stroebel et al., 2018*). Indeed, the fast component of the amplitude is reduced in *Ppt1*$^{-/-}$ neurons (*Figure 3B*), indicating a functional decrease in the contribution of GluN2A to NMDAR-mediated EPSCs in these cells. Moreover, analysis of the weighted decay time constant (*Figure 3C*) suggests a larger contribution of GluN2B to the overall NMDAR-EPSC in

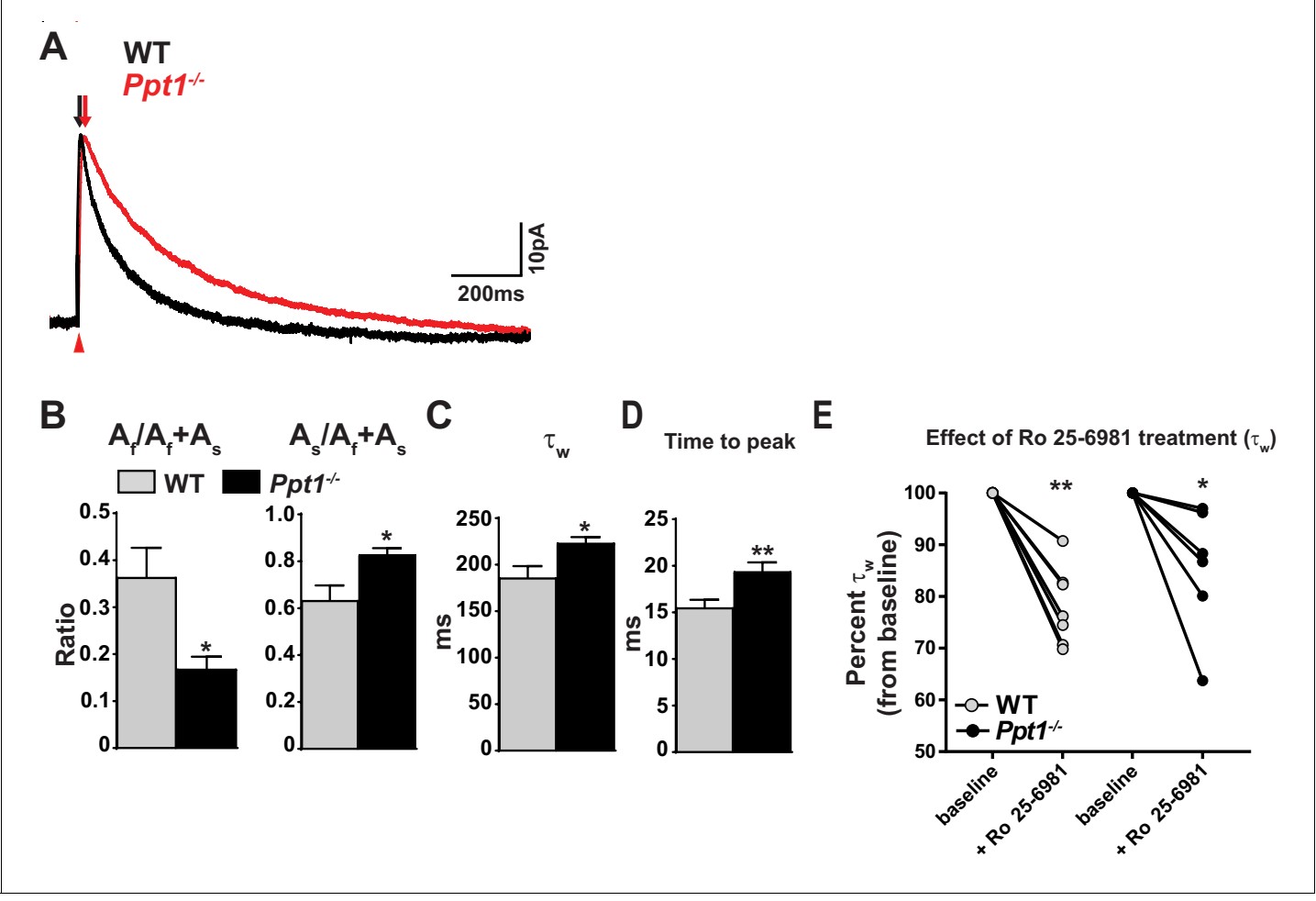

**Figure 3.** NMDAR-mediated EPSCs are altered in $Ppt1^{-/-}$ visual cortex. (**A**) Representative traces of amplitude-scaled NMDAR-EPSCs recorded from pyramidal neurons in layer II/III of the visual cortex (V1) of WT and $Ppt1^{-/-}$ mice. Black and red arrows above traces indicate EPSC rise time for WT and $Ppt1^{-/-}$ responses, respectively. Red arrow below traces indicates onset of evoked stimulus. Neurons were voltage clamped at +50 mV and NMDAR-EPSCs evoked in layer IV. Neurons were voltage clamped at +50 mV and NMDAR-EPSCs evoked in layer IV. (**B**) Quantification of the ratio of the amplitude (**A**) of the fast component, $A_f/A_f + A_s$, and $A_s/A_f + A_s$ derived from fitting the decay phase of the evoked NMDAR-EPSCs with the double exponential function: $Y(t) = A_f*e^{-t/t_{fast}} + A_s*e^{-t/t_{slow}}$. (**C**) Quantification of the weighted decay constant, $\tau_w$ derived from fitting the decay phase of the amplitude-scaled evoked NMDAR-EPSCs with the double exponential function: $Y(t) = A_f*e^{-t/t_{fast}} + A_s*e^{-t/t_{slow}}$. (**D**) Quantification of the NMDAR-EPSC time to peak amplitude. (**E**) Percent change in $\tau_w$ following bath application of Ro 25–6981 (3 µM, 30 min) for each cell in WT and $Ppt1^{-/-}$ neurons. For experiments in **Figure 3A–D**, $Ppt1^{-/-}$ and WT were compared (n = 8 cells, four mice (WT); n = 8 cells, five mice ($Ppt1^{-/-}$)) using t-test and the significance was indicated as follows: *p<0.05, and **p<0.01. For experiments in **Figure 3E**, the change in $\tau_w$ from baseline induced by Ro 25–6981 were compared in WT and $Ppt1^{-/-}$ neurons (n = 7 cells, four mice (WT); n = 6 cells, four mice ($Ppt1^{-/-}$)) using repeated measures two-way ANOVA followed by Tukey's post-hoc test and significance was indicated as follows: *p<0.05, and **p<0.01 vs. baseline. Error bars represent s.e.m.

DOI: https://doi.org/10.7554/eLife.40316.005

$Ppt1^{-/-}$ cells (**Stocca and Vicini, 1998**; **Vicini et al., 1998**). Thus, our findings suggest an enhanced incorporation of triheteromeric NMDARs at $Ppt1^{-/-}$ synapse, and corroborate our biochemical findings (see **Discussion**). Collectively, our data indicate a functionally immature NMDAR phenotype in $Ppt1^{-/-}$ layer II/III visual cortical neurons.

## Dendritic spine morphology is immature in $Ppt1^{-/-}$ visual cortex

The morphology of dendritic spines is dynamic and modified by synaptic plasticity (**Engert and Bonhoeffer, 1999**; **Parnass et al., 2000**; **Yuste and Bonhoeffer, 2001**; **Matsuzaki et al., 2004**). During visual cortical development that is concomitant with the GluN2B to GluN2A switch, dendritic spine morphology undergoes robust structural plasticity at excitatory synapses. Dendritic spines contribute

to experience-dependent synaptic plasticity via the generation, maturation, and long-term stabilization of spines, ultimately giving rise to established synaptic circuits. Typically, by P33, dendritic spines begin to demonstrate a reduction in turnover and an increase in mushroom-type spines, indicating synaptic maturity. Importantly, dendritic spines are morphologically disrupted in many neurodevelopmental disorders, typically skewing toward an immature phenotype (*Purpura, 1979*; *Irwin et al., 2001*; *Penzes et al., 2011*).

We hypothesized that dendritic spine morphology is immature or disrupted in *Ppt1⁻/⁻* neurons, particularly given that GluN2A subunit incorporation is disrupted in vivo. Thus, we used in utero electroporation to sparsely label layer II/III cortical neurons in the visual cortex using a GFP construct (*Matsuda and Cepko, 2004*). GFP-expressing cells from WT and *Ppt1⁻/⁻* animals were imaged for detailed analysis of dendritic spine morphology (spine length, spine volume, and spine head volume) at P33, a time point when dendritic spine morphology is typically considered mature and GluN2A is reduced at *Ppt1⁻/⁻* synapses.

We analyzed dendritic spine characteristics of GFP-expressing cells (procedure schematized in *Figure 4A*) from WT and *Ppt1⁻/⁻* visual cortex (*Figure 4B*) using the Imaris software (Bitplane). While WT neurons exhibited mushroom-type spine morphology with high-volume spine heads (*Figure 4C*, arrows), *Ppt1⁻/⁻* neurons showed longer, filopodial protrusions or stubby spines (*Figure 4C*, arrowheads). Quantification of spine length and spine volume demonstrated that *Ppt1⁻/⁻* spines were longer and less voluminous compared to WT (*Figure 4D–E*). Further, the volume of dendritic spine heads was reduced in *Ppt1⁻/⁻* neurons (*Figure 4E*, inset). Interestingly, dendritic spine density was significantly increased in *Ppt1⁻/⁻* neurons, signifying dysregulated synapse formation or refinement in the *Ppt1⁻/⁻* brain (*Figure 4F*). These data indicate that dendritic spine morphology is disrupted in the developing CLN1 visual cortex, corresponding with the finding that NMDAR composition is immature at P33 and suggesting a reduced ability to compartmentalize calcium and other localized biochemical signals in CLN1.

## NMDAR subunit composition and dendritic spine morphology are also immature in *Ppt1⁻/⁻* primary cortical neurons

The GluN2B to GluN2A switch and maturation of dendritic spine characteristics in WT primary neurons has been previously demonstrated (*Williams et al., 1993*; *Zhong et al., 1994*; *Papa et al., 1995*). We established that the developmental switch from GluN2B- to GluN2A-containing NMDARs and dendritic spine morphology are impaired in the *Ppt1⁻/⁻* mouse brain. To understand these mechanisms more comprehensively and examine protein palmitoylation more directly, we used dissociated neuronal cultures. First, we analyzed these developmental events in WT and *Ppt1⁻/⁻* primary cortical neurons to determine whether the biochemical and structural features of disease are recapitulated in vitro.

We collected lysates from cultured cortical neurons for 7, 10, or 18 days in vitro (DIV 7, 10, or 18) harvested and performed immunoblot analyses for markers of immature (GluN2B) or mature (GluN2A, PSD-95) excitatory synapses. Expression of GluN2B clearly preceded that of mature synaptic markers, peaking in both WT and *Ppt1⁻/⁻* neurons at DIV10 and decreasing slightly thereafter (*Figure 5A*). In contrast, levels of both GluN2A and PSD-95 remained low until DIV18, at which point expression was robust (*Figure 5B,C*). Importantly, GluN2A, PSD-95, and GluN2A/GluN2B ratio levels showed reductions in *Ppt1⁻/⁻* neurons compared to WT at DIV18, indicating that the biochemical phenotype is recapitulated to an extent in vitro (*Figure 5B–D*).

To analyze dendritic spine morphology, we transfected primary cortical neurons from fetal WT and *Ppt1⁻/⁻* mice with the GFP construct as above (*Matsuda and Cepko, 2004*) and cultured until DIV 15 or 20, then performed live cell imaging (*Figure 6A*). We measured dendritic spine length and volume in transfected cells using the Imaris software (Bitplane). At both DIV 15 and 20, we observed a significant alterations in the dendritic spine length (*Figure 6B–E*). The distribution of spine length in *Ppt1⁻/⁻* neurons at DIV15 significantly shifted toward longer protrusions as compared to WT cells (*Figure 6B*). The averaged spine length was also robustly increased in *Ppt1⁻/⁻* neurons (*Figure 6C*). Similar changes were also present at DIV20 (*Figure 6D–E*). Next, we analyzed differences in dendritic spine volume (*Figure 6F–I*). *Ppt1⁻/⁻* neurons exhibit a significant reduction in the percentage of spines with volumes greater than ~0.2 μm³ at both DIV15 (*Figure 6F*) and DIV20 (*Figure 6H*). The averaged spine volume was also reduced in *Ppt1⁻/⁻* neurons at both DIV15 (*Figure 6G*) and DIV20 (*Figure 6I*). As observed in vivo, *Ppt1⁻/⁻* neurons showed an increase in the dendritic spine density at

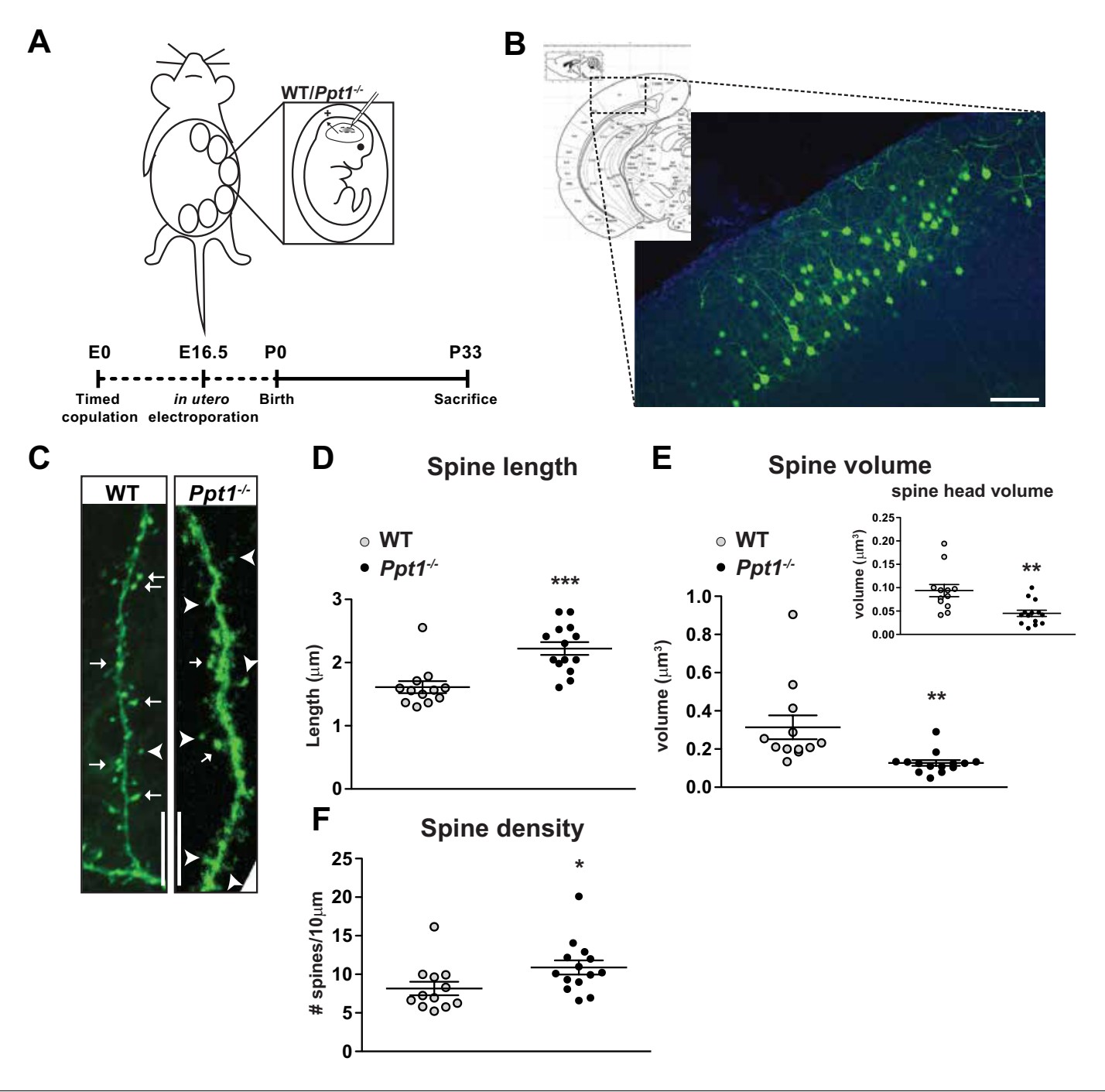

**Figure 4.** Dendritic spine morphology is immature in *Ppt1⁻ᐟ⁻* layer II/III visual cortical neurons. (**A**) Schematic of in utero electroporation procedure and timeline (bottom) (**B**) coronal diagram from Paxinos' mouse brain atlas demonstrating areas of visual cortex (left) and representative low-magnification (10x) confocal image of a successfully transfected group of layer II/III neurons in visual cortex (right). Scale bar = 100 μm. (**C**) Representative confocal images of GFP-transfected dendritic segments from WT and *Ppt1⁻ᐟ⁻* neurons at P33. Arrows mark mature, mushroom-type spines; arrowheads mark thin, filopodial spines or stubby, headless spines. Scale bar = 10 μm. (**D**) Semi-automated quantification of dendritic spine length in WT and *Ppt1⁻ᐟ⁻* visual cortical neurons at P33. (**E**) Semi-automated quantification of dendritic spine volume and spine head volume (inset) in WT and *Ppt1⁻ᐟ⁻* visual cortical neurons at P33. (**F**) Semi-automated quantification of dendritic spine density per 10 μm of dendrite in WT and *Ppt1⁻ᐟ⁻* visual cortical neurons at P33. For experiments in *Figure 4*, WT and *Ppt1⁻ᐟ⁻* were compared (n = 3–4 cells/animal, three animals/group) using t-test and the significance was indicated as follows: *p<0.05, **p<0.01, ***p<0.001, *Ppt1⁻ᐟ⁻* vs. WT. Error bars represent s.e.m.

DOI: https://doi.org/10.7554/eLife.40316.006

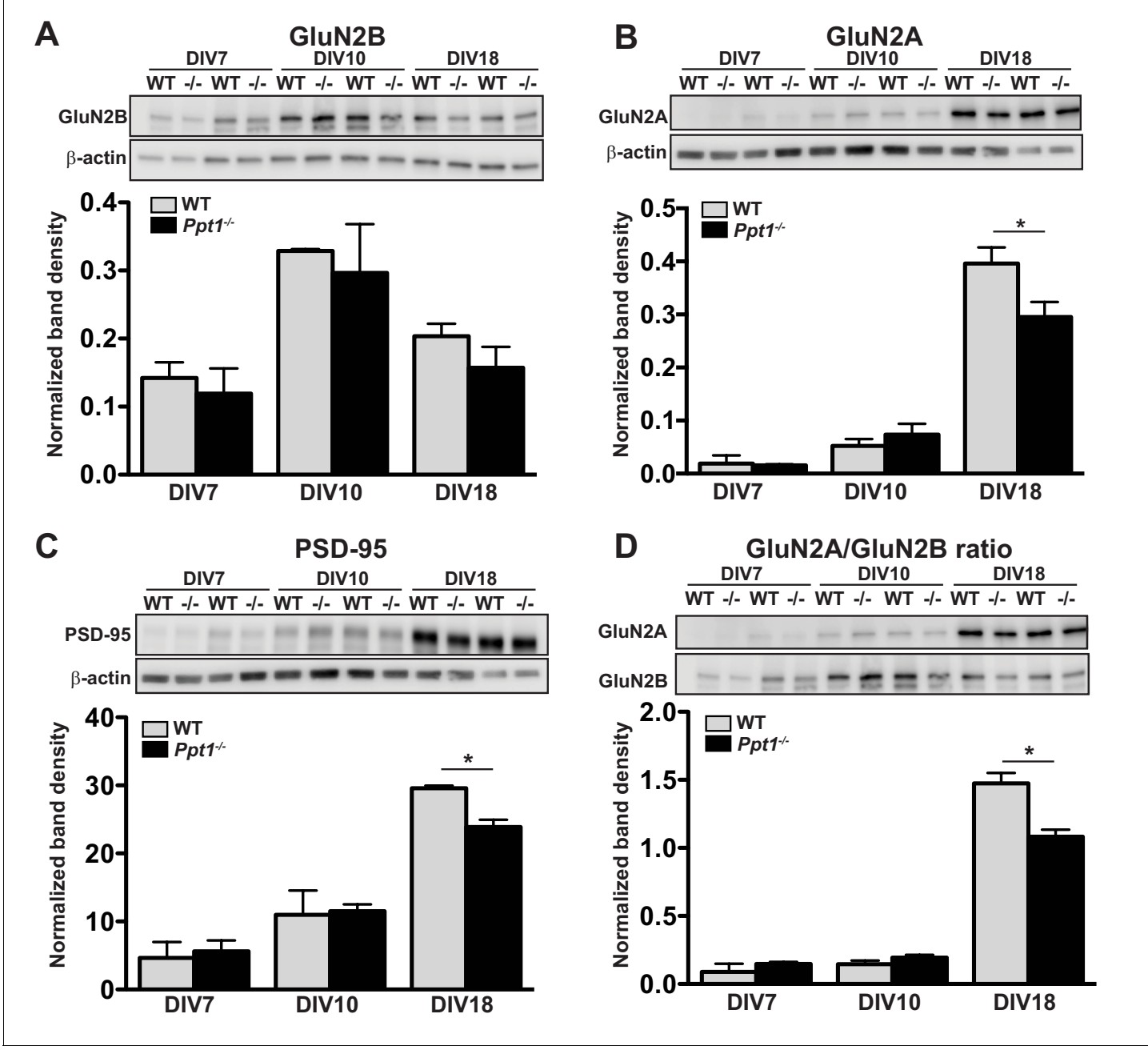

**Figure 5.** GluN2B to GluN2A NMDAR switch and *Ppt1⁻/⁻*-induced synaptic deficits are recapitulated in primary cortical neurons. (**A**) Representative immunoblot (top) and quantification of GluN2B levels in WT and *Ppt1⁻/⁻* neurons at DIV7, 10, and 18. (**B**) Representative immunoblot (top) and quantification of GluN2A levels (bottom) in WT and *Ppt1⁻/⁻* neurons at DIV7, 10, and 18. (**C**) Representative immunoblot (top) and quantification of PSD-95 levels (bottom) in WT and *Ppt1⁻/⁻* neurons at DIV7, 10, and 18. (**D**) Representative immunoblot (top) and quantification of the GluN2A/2B ratio (bottom) in WT and *Ppt1⁻/⁻* neurons at DIV7, 10, and 18. For all experiments in *Figure 5*, *Ppt1⁻/⁻* and WT were compared (n = 2 independent experiments with two repetitions/group) at each time point using t-test and the significance indicated as follows: *p<0.05 where indicated. Error bars represent s.e.m.

DOI: https://doi.org/10.7554/eLife.40316.007

DIV15 (*Figure 6J*) and DIV20 (*Figure 6K*), again suggesting aberrant synapse formation, a failure of synaptic pruning, or both in *Ppt1⁻/⁻* neurons. Together, these data demonstrate that *Ppt1⁻/⁻* neurons in culture give rise to morphologically immature dendritic spines and corroborate our in vivo findings.

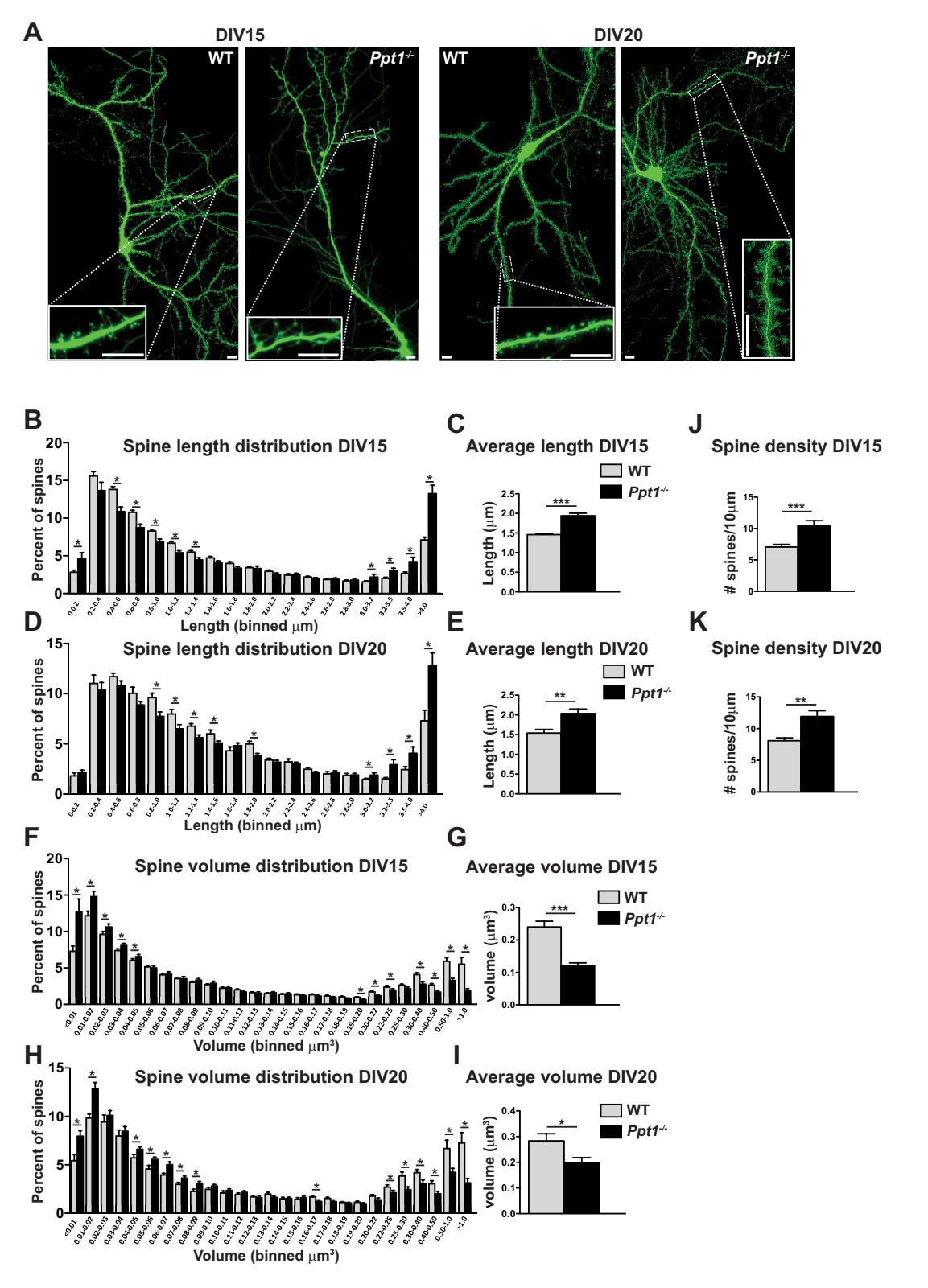

**Figure 6.** Dendritic spine morphology is immature in *Ppt1⁻/⁻* neurons in vitro. (**A**) Representative composite confocal images of live DIV15 (left) and DIV20 (right) GFP-transfected, cultured WT and *Ppt1⁻/⁻* neurons. Insets represent dendrite segments within dotted line. Scale bar = 10 μm. (**B**) Quantification of dendritic spine length in WT and *Ppt1⁻/⁻* neurons at DIV15. Spine length is binned into 19 discrete groups from 0 - > 4 μm. (**C**) Mean length of all spines in cultured WT and *Ppt1⁻/⁻* neurons at DIV15. (**D**) Quantification of dendritic spine length in WT and *Ppt1⁻/⁻* neurons at DIV20. Spine

*Figure 6 continued on next page*

Figure 6 continued

length is binned into 19 discrete groups from 0 - > 4 µm. (E) Mean length of all spines in cultured WT and $Ppt1^{-/-}$ neurons at DIV20. (F) Semi-automated quantification of dendritic spine volume in WT and $Ppt1^{-/-}$ cultured neurons at DIV15. Spine volume is binned into 27 discrete groups form 0 - > 1 µm³. (G) Mean volume of all spines in cultured WT and $Ppt1^{-/-}$ neurons at DIV15. (H) Semi-automated quantification of dendritic spine volume in WT and $Ppt1^{-/-}$ cultured neurons at DIV20. Spine volume is binned into 27 discrete groups form 0 - > 1 µm³. (I) Mean volume of all spines in cultured WT and $Ppt1^{-/-}$ neurons at DIV20. (J) Semi-automated quantification of dendritic spine density per 10 µm of dendrite in WT and $Ppt1^{-/-}$ cultured neurons at DIV15. (K) Semi-automated quantification of dendritic spine density per 10 µm of dendrite in WT and $Ppt1^{-/-}$ cultured neurons at DIV20. For experiments in **Figure 6**, $Ppt1^{-/-}$ and WT were compared (For DIV15: n = 4–5 neurons/group, three-independent experiments, WT = 21,514 spines; $Ppt1^{-/-}$ = 18,013 spines. For DIV20: n = 3 neurons/group, two-independent experiments, WT = 11,335 spines; $Ppt1^{-/-}$ = 9958 spines) using t-test (within bin in the case of distribution graphs) and the significance was indicated as follows: *p<0.05, **p<0.05, ***p<0.001 where indicated. Error bars represent s.e.m.

DOI: https://doi.org/10.7554/eLife.40316.008

## Calcium imaging reveals extrasynaptic calcium dynamics in $Ppt1^{-/-}$ neurons

Intracellular calcium dynamics, compartmentalization, and signaling play a critical role in synaptic transmission and plasticity. These properties are altered by glutamate receptor composition and location (*Lau and Zukin, 2007*; *Hardingham and Bading, 2010*; *Paoletti et al., 2013*). GluN2B-containing NMDARs maintain a prolonged open conformation compared to GluN2A-containing receptors, allowing increased calcium entry per synaptic event (*Sobczyk et al., 2005*). Moreover, previous studies indicate that GluN2A-containing NMDARs are generally inserted in the PSD, whereas GluN2B-containing NMDARs are localized extrasynaptically and associated with SAP102 (*Tovar and Westbrook, 1999*; *Townsend et al., 2003*; *van Zundert et al., 2004*; *Washbourne et al., 2004*; *Groc et al., 2007*; *Elias et al., 2008*; *Martel et al., 2009*). To determine more directly the effects of our biochemical and electrophysiological findings on calcium dynamics, we analyzed calcium signals in WT and $Ppt1^{-/-}$ neurons transfected with the genetically encoded calcium sensor, GCaMP3 (*Tian et al., 2009*).

While WT neurons exhibited primarily compartmentalized calcium signals that were restricted to individual spines (**Figure 7A–C**, left, see **Video 1**), $Ppt1^{-/-}$ neurons demonstrated diffuse calcium influxes that spread through the dendritic shaft (**Figure 7A–C**, right, see **Video 2**). These extrasynaptic transients appear rarely in WT cells (**Figure 7**, see Videos). To analyze the calcium dynamics in more detail, measurements of $\Delta F/F_0$ were made for each dendritic segment, from each cell over the course of the captured videos (see Materials and methods). Multiple transients from the same synaptic site are shown as a heat map of $\Delta F/F_0$ measurements and they are largely consistent across time in both WT and $Ppt1^{-/-}$ neurons (**Figure 7B**). Further, plotting of the averaged $\Delta F/F_0$ transients at an individual synaptic site demonstrates that local fluorescence increases in WT cells are confined to a short distance from the peak $\Delta F/F_0$ at synaptic sites (**Figure 7B and C**, left), while those of $Ppt1^{-/-}$ neurons diffuse longer distances within the dendrite (**Figure 7B and C**, right). To quantitatively compare these properties, we performed measurements of area under the curve (AUC) and calcium diffusion distance (see shaded region in **Figure 7C**) for each synaptic site from WT and $Ppt1^{-/-}$ neurons. These analyses revealed a robust increase in both the AUC (**Figure 7D**) and the calcium diffusion distance (**Figure 7E**) in $Ppt1^{-/-}$ neurons compared to WT. Furthermore, performing correlation analysis of calcium events across time (see Materials and methods) within a given neuron demonstrates that calcium influxes are more synchronous (increased correlation coefficient) in $Ppt1^{-/-}$ neurons compared to WT (**Figure 7F**). This result may involve mechanisms underlying synaptic cluster plasticity, including synaptic integration via translational activation influenced by excessive $Ca^{2+}$ entry (*Govindarajan et al., 2006*), enhanced biochemical crosstalk between synapses by, for example, small GTPases (which are generally palmitoylated proteins) (*Harvey et al., 2008*), or direct cooperative multi-synaptic $Ca^{2+}$ signaling (*Weber et al., 2016*) in $Ppt1^{-/-}$ neurons. Together, these data indicate that calcium entry and dispersion are enhanced at $Ppt1^{-/-}$ synapses in vitro.

These data are in line with our biochemical and electrophysiological findings and suggest that GluN2B-containing NMDARs mediate the observed calcium signals. To further test this possibility, we next treated WT and $Ppt1^{-/-}$ neurons with Ro 25–6981 (1 µM, added in imaging medium following 2.5 min imaging at baseline) and performed calcium imaging. Ro 25–6981 had virtually no effect on calcium signals recorded from WT cells (**Figure 7G–I**, see **Video 3**). In contrast, $Ppt1^{-/-}$ neurons

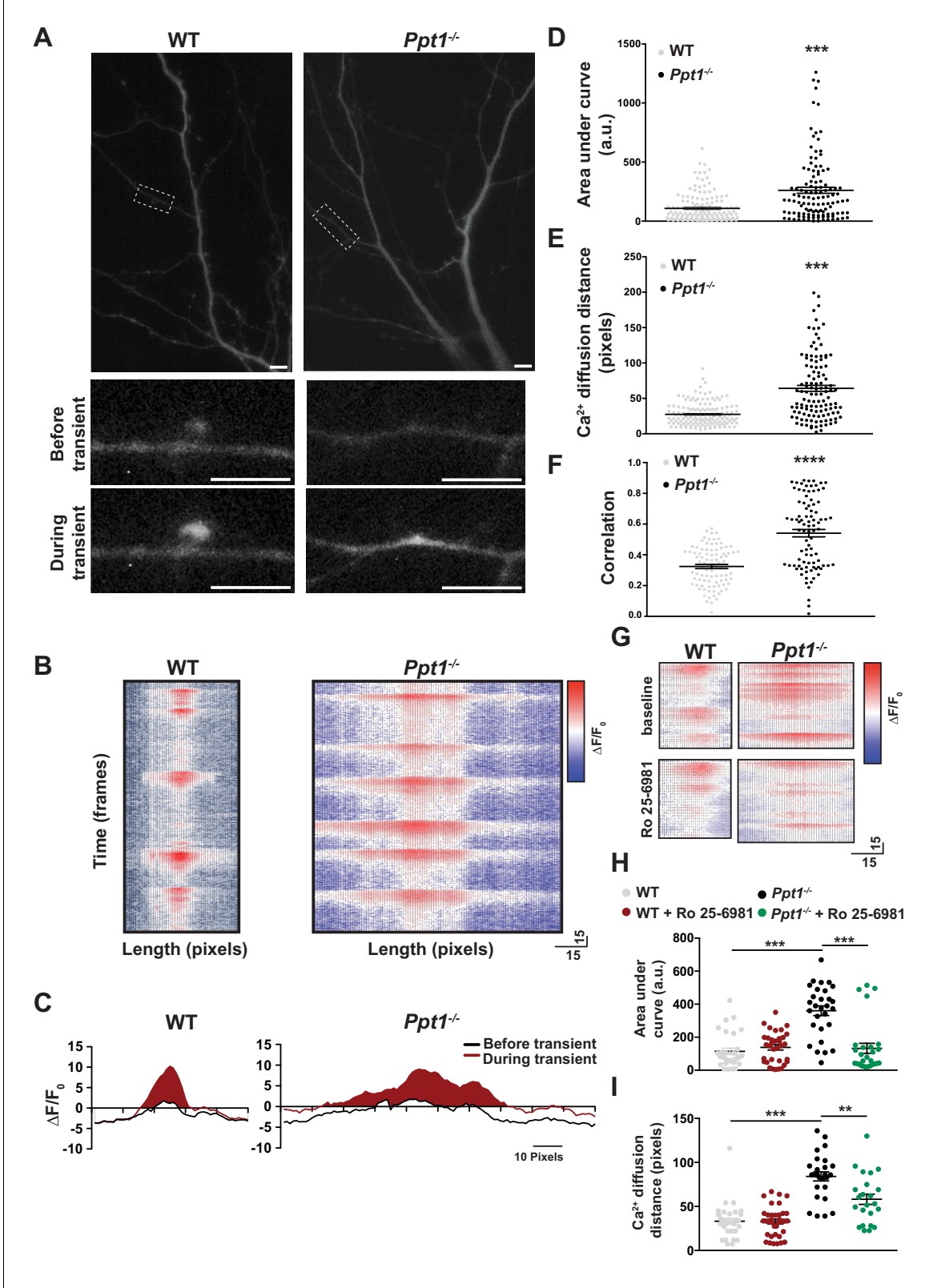

**Figure 7.** Calcium imaging reveals extrasynaptic calcium dynamics in *Ppt1*⁻/⁻ neurons. DIV16-18, WT and *Ppt1*⁻/⁻ cortical neurons transfected with GCaMP3 and imaged in the absence of $Mg^{2+}$ for 5 min. (**A**) Single frames from *Videos 1* and *2* of WT (left, note that cell is rotated 90° from *Video 1*) and *Ppt1*⁻/⁻ (right) cultured neurons. Dendritic segments within the dotted-lines represent zoomed-in images of a single spine (left, WT) or dendritic shaft segment (right, *Ppt1*⁻/⁻) at baseline (top) and active (bottom) states. Scale = 10 µm. (**B**) Representative heat maps of $\Delta F/F_0$ values at one synaptic

*Figure 7 continued on next page*

*Figure 7 continued*

site from WT (left) and *Ppt1⁻/⁻* (right) dendrite segments during a portion the imaging session (350 frames, 50 s). (C) Representative averaged $\Delta F/F_0$ responses at one synaptic site from WT (left) and *Ppt1⁻/⁻* (right) neurons. Area under the curve represents calcium influx and is shaded in red. (D) Quantification of calcium transient area under the curve WT and *Ppt1⁻/⁻* neurons. (E) Quantification of calcium transient diffusion distance from WT and *Ppt1⁻/⁻* neurons. (F) Quantification of average correlation coefficient (synaptic synchrony) across time between sites of synaptic activity in WT and *Ppt1⁻/⁻* neurons. (G) Representative heat maps of $\Delta F/F_0$ values at one synaptic site from WT (left) and *Ppt1⁻/⁻* (right) dendrite segments before (top) and after (bottom) treatment with Ro 25–6981 (130 frames, 18 s). (H) Quantification of calcium transient area under the curve WT and *Ppt1⁻/⁻* neurons before and after treatment with Ro 25–6981. (I) Quantification of calcium transient diffusion distance from WT and *Ppt1⁻/⁻* neurons before and after treatment with Ro 25–6981. For experiments in *Figure 7D–E*, *Ppt1⁻/⁻* and WT were compared (n = 185 synaptic sites (WT), n = 131 synaptic sites (*Ppt1⁻/⁻*), three neurons/group, three individual experiments) by t-test and the significance was indicated as follows: ***p<0.001 vs. WT by t-test. For experiments in *Figure 7F*, *Ppt1⁻/⁻* and WT were compared (n = 100 synaptic sites (WT), n = 100 synaptic sites (*Ppt1⁻/⁻*); three neurons/group, two individual experiments) by t-test and the significance was indicated as follows: ***p<0.001 vs. WT by t-test. For experiments in *Figure 7H–I*, *Ppt1⁻/⁻* and WT were compared (n = 25 synaptic sites (WT), n = 28 synaptic sites (*Ppt1⁻/⁻*); three neurons/group, two individual experiments) by t-test and the significance was indicated as follows: ***p<0.001 vs. WT by t-test. 65 pixels is representative of 10 µm. Error bars represent s.e.m.

DOI: https://doi.org/10.7554/eLife.40316.009

treated with Ro 25–6981 showed a reduction in dendritic calcium influxes within shafts, while few residual, compartmentalized transients persisted (*Figure 7G–I*, see *Video 4*). Quantitatively, both AUC (*Figure 7H*) and calcium diffusion (*Figure 7I*) distance were rescued to WT levels following Ro 25–6981 treatment of *Ppt1⁻/⁻* neurons. Together, these data suggest that *Ppt1⁻/⁻* neurons have extrasynaptic calcium signaling compared to WT that is sensitive to GluN2B-NMDAR blockade.

## *Ppt1⁻/⁻* cultured neurons show enhanced vulnerability to NMDA-mediated excitotoxicity

GluN2B-predominant NMDARs are implicated in enhanced neuronal susceptibility to NMDA-mediated neuronal death (*Martel et al., 2009*; *Martel et al., 2012*). Our results from biochemical, electrophysiological, and live-imaging analyses indicate decreased GluN2A/2B ratio suggesting an intriguing possibility that *Ppt1⁻/⁻* neurons are more vulnerable to excitotoxicity (*Finn et al., 2012*). Therefore, we treated WT and *Ppt1⁻/⁻* cultured neurons with NMDA (varying doses, 10–300 µm) and glycine (1–30 µm, always in 1:10 ratio with NMDA) for 2 hr and assayed cell viability 24 hr later using the PrestoBlue reagent (ThermoFisher Scientific) (*Figure 8A*). As expected, both WT and *Ppt1⁻/⁻* neurons demonstrated dose-dependent reductions in cell viability in response to increasing concentrations of NMDA/glycine (*Figure 8B*). Importantly, *Ppt1⁻/⁻* neurons were more vulnerable to NMDA insult, as exposure to 10 µM NMDA was sufficient to reduce cell viability significantly in *Ppt1⁻/⁻* neurons but not WT cells (WT = 93 ± 4.1%; *Ppt1⁻/⁻* = 76 ± 3.5%; *p=0.046; *Figure 8B*). Further, at 100 µM NMDA, WT neuron viability decreased by 35%, while *Ppt1⁻/⁻* neuron viability was reduced significantly further, by 58% (WT = 65 ± 1.8%; *Ppt1⁻/⁻* = 42 ± 4.5%; **p=0.0043; *Figure 8B*). At 300 µM NMDA treatment this effect plateaued, as cell viability between WT and *Ppt1⁻/⁻* neurons was comparable (*Figure 8B*). These results indicate *Ppt1⁻/⁻* neurons are more vulnerable to excitotoxicity and are consistent with our calcium imaging data that demonstrated the predominance of extrasynaptic, GluN2B-mediated NMDAR activity.

## Palmitoylation inhibitors rescue enhanced vulnerability to NMDA-mediated excitotoxicity in *Ppt1⁻/⁻* cultured neurons

We next asked whether this enhanced vulnerability to excitotoxicity results from hyperpalmitoylation of neuronal substrates, and if it can be corrected by balancing the level of synaptic protein palmitoylation/depalmitoylation. First, we found that 77% of cultured *Ppt1⁻/⁻* neurons accumulate ALs spontaneously at DIV18-20 (*Figure 9A and B*). In agreement, an

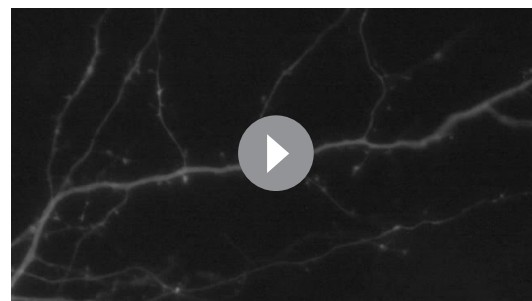

**Video 1.** Spontaneous calcium activity in DIV16-18 WT neuron. Representative video of spontaneous neuronal calcium activity in a WT cultured neuron at DIV16-18.
DOI: https://doi.org/10.7554/eLife.40316.010

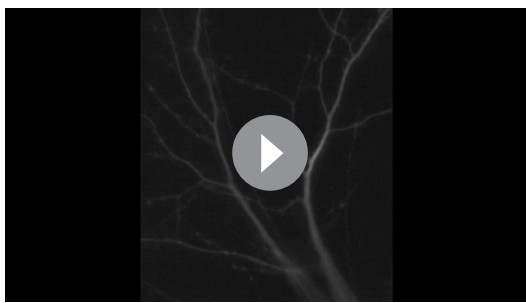

**Video 2.** Spontaneous calcium activity in DIV16-18 *Ppt1*<sup>-/-</sup> neuron. Representative video of spontaneous neuronal calcium activity in a *Ppt1*<sup>-/-</sup> cultured neuron at DIV16-18.
DOI: https://doi.org/10.7554/eLife.40316.011

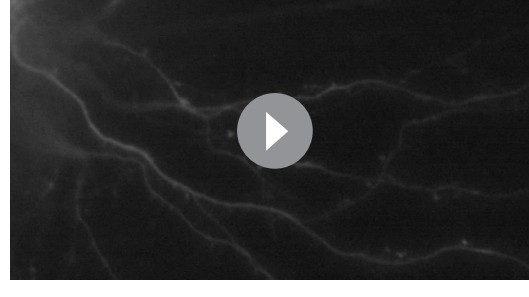

**Video 3.** Spontaneous calcium activity in DIV16-18 WT neuron before and after treatment with Ro 25–6981. Representative video of spontaneous neuronal calcium activity in a WT cultured neuron at DIV16-18 prior to, and following, bath application of Ro 25–6981 (1 μM, after 30 s).
DOI: https://doi.org/10.7554/eLife.40316.012

immunostaining for lysosomal-associated membrane protein-2 (LAMP-2) showed colocalization of the lysosomal marker with ALs in *Ppt1*<sup>-/-</sup> but not WT neurons (vehicle treatment in *Figure 9B*, *Figure 9—figure supplement 1*). Further, lysosomes appeared swollen in vehicle-treated *Ppt1*<sup>-/-</sup> neurons (see arrows in *Figure 9B*, *Figure 9—figure supplement 1D–F*). Treatment with the palmitoylation inhibitors, 2-bromopalmitate (2 BP, 1 μM, 7 day treatment) and cerulenin (1 μM, 7 day treatment) reduced the percentage of AL-positive neurons (*Figure 9C*) and the area occupied with ALs per neuron (*Figure 9D*). Further, the mean lysosomal size also normalized in *Ppt1*<sup>-/-</sup> neurons when these cells were treated with 2 BP or cerulenin (*Figure 9E*).

To examine the efficacy of these compounds in preventing NMDA-mediated toxicity, we pretreated a subset of neurons with the same palmitoylation inhibitors, 2 BP (1 μM, DIV12-18) and cerulenin (1 μM, DIV12-18) prior to treatment with NMDA and glycine. Notably, pretreatment with both 2 BP and cerulenin improved cell viability of *Ppt1*<sup>-/-</sup> neurons to that of WT following excitotoxicity induction, while the chronic low-dose treatment alone had no effect on neuronal viability (*Figure 9F*). These results indicate at least two features of *Ppt1*<sup>-/-</sup> neurons, accumulation of proteolipid materials and a higher vulnerability to excitotoxicity, can be mitigated by correcting a balance between palmitoylation and depalmitoylation.

## Palmitoylation inhibitor treatment improves pathological calcium dynamics in *Ppt1*<sup>-/-</sup> neurons

To determine whether palmitoylation inhibitor treatment had a functional effect on the calcium dynamics in *Ppt1*<sup>-/-</sup> neurons, we treated a subset of *Ppt1*<sup>-/-</sup> cells from DIV12-18 with 2 BP (1 μM) or cerulenin (1 μM) before imaging under the same conditions described for *Figure 7* (see *Videos 5* and *6*). Notably, treatment with both 2 BP and cerulenin decreased the AUCs of specified synapses compared to untreated *Ppt1*<sup>-/-</sup> cells (see *Video 7*), nearly to WT levels (*Figure 10A*), indicating more compartmentalized calcium influx. In fact, the morphology of treated *Ppt1*<sup>-/-</sup> neurites appeared more mature (see *Videos 5–7*). However, the AUC of cerulenin-treated *Ppt1*<sup>-/-</sup> neurons was still significantly increased compared to WT, indicating a partial rescue of phenotype (*Figure 10A*). We also observed similar changes for the calcium diffusion distance at

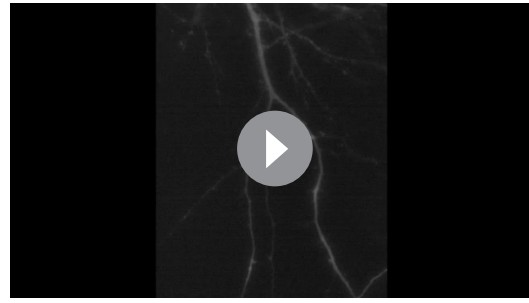

**Video 4.** Spontaneous calcium activity in DIV16-18 *Ppt1*<sup>-/-</sup> neuron before and after treatment with Ro 25–6981. Representative video of spontaneous neuronal calcium activity in a *Ppt1*<sup>-/-</sup> cultured neuron at DIV16-18 prior to, and following, bath application of Ro 25–6981 (1 μM, after 31 s).
DOI: https://doi.org/10.7554/eLife.40316.013

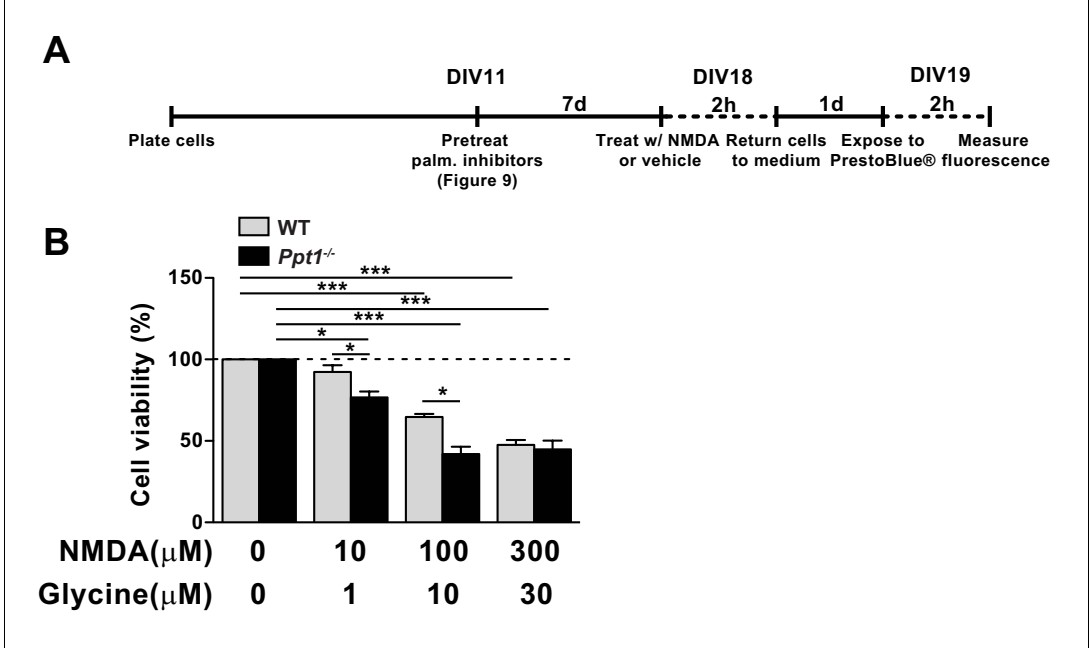

**Figure 8.** $Ppt1^{-/-}$ cultured neurons show enhanced vulnerability to NMDA-mediated excitotoxicity. (**A**) Schematic of cellular toxicity experimental design. Briefly, neurons were grown to DIV11, treated with vehicle of palmitoylation inhibitors for 7 days (every 48 hr) and neuronal viability was measured by PrestoBlue cellular viability assay following exposure (2 hr exposure, 22 hr incubation in medium) to NMDA and glycine. (**B**) Quantification of cellular viability in WT and $Ppt1^{-/-}$ neurons at DIV19 treated with increasing concentrations of NMDA and glycine (10/1, 100/10, and 300/30 µM). $Ppt1^{-/-}$ and WT were compared (n = 4 independent experiments, in duplicate) by two-way ANOVA followed by Tukey's post hoc test and significance was indicated as follows: *p<0.05 and ***p<0.001 where indicated. Error bars represent s.e.m.

DOI: https://doi.org/10.7554/eLife.40316.014

synaptic sites, as groups followed the order: $Ppt1^{-/-} > Ppt1^{-/-} +$ cerulenin $\geq Ppt1^{-/-} +$ 2 BP=WT. These results further confirm the efficacy of palmitoylation inhibitor treatment (**Figure 10B**). Further, calcium transient frequency was higher in $Ppt1^{-/-}$ cells than in WT but was lowest in $Ppt1^{-/-}$ neurons treated with 2 BP or cerulenin (**Figure 10C and D**). It is plausible that chronic palmitoylation inhibitor treatment caused dissipation of synaptic proteins, including NMDARs (**El-Husseini et al., 2002**; **Li et al., 2003**), thereby reducing transient frequency.

## Palmitoylation inhibitors rescue Fyn kinase and GluN2B hyperpalmitoylation in $Ppt1^{-/-}$ neurons

Finally, we directly examined the palmitoylation state of neuronal proteins to gain insight into the mechanisms by which hyperpalmitoylation of neuronal substrates may lead to NMDA-mediated excitotoxicity in $Ppt1^{-/-}$ neurons. We also asked whether palmitoylation inhibitors can correct these abnormalities. We employed a modified acyl-biotin exchange procedure (**Drisdel and Green, 2004**), termed the APEGS assay (acyl-PEGyl exchange gel-shift) (**Yokoi et al., 2016**). The APEGS assay effectively tags the palmitoylation sites of neuronal substrates with a 5 kDa polyethylene glycol (PEG) polymer, causing a molecular weight-dependent gel shift in immunoblot analyses. Thus, we quantitatively analyzed the palmitoylated fraction of synaptic proteins and palmitoylated signaling molecules that may influence NMDAR function.

To test the feasibility of the APEGS assay in our primary cortical neuronal cultures, we collected lysates at DIV18 from WT, $Ppt1^{-/-}$, and palmitoylation inhibitor-treated (2 BP or cerulenin, DIV12-18, 1 µm) neurons and examined two palmitoylated proteins, PSD-95 and Fyn, which have been successfully quantified using this method (**Yokoi et al., 2016**).

We examined the palmitoylation state of PSD-95 at baseline and in response to palmitoylation inhibitor treatment (**Figure 11A**). As we found in our initial immunoblotting analyses of the homogenates derived from cortical tissues (**Figure 2**) and neuronal cultures (**Figure 5**), $Ppt1^{-/-}$ neurons had lower amounts of total PSD-95 protein than WT, as evidenced by decreased overall band density

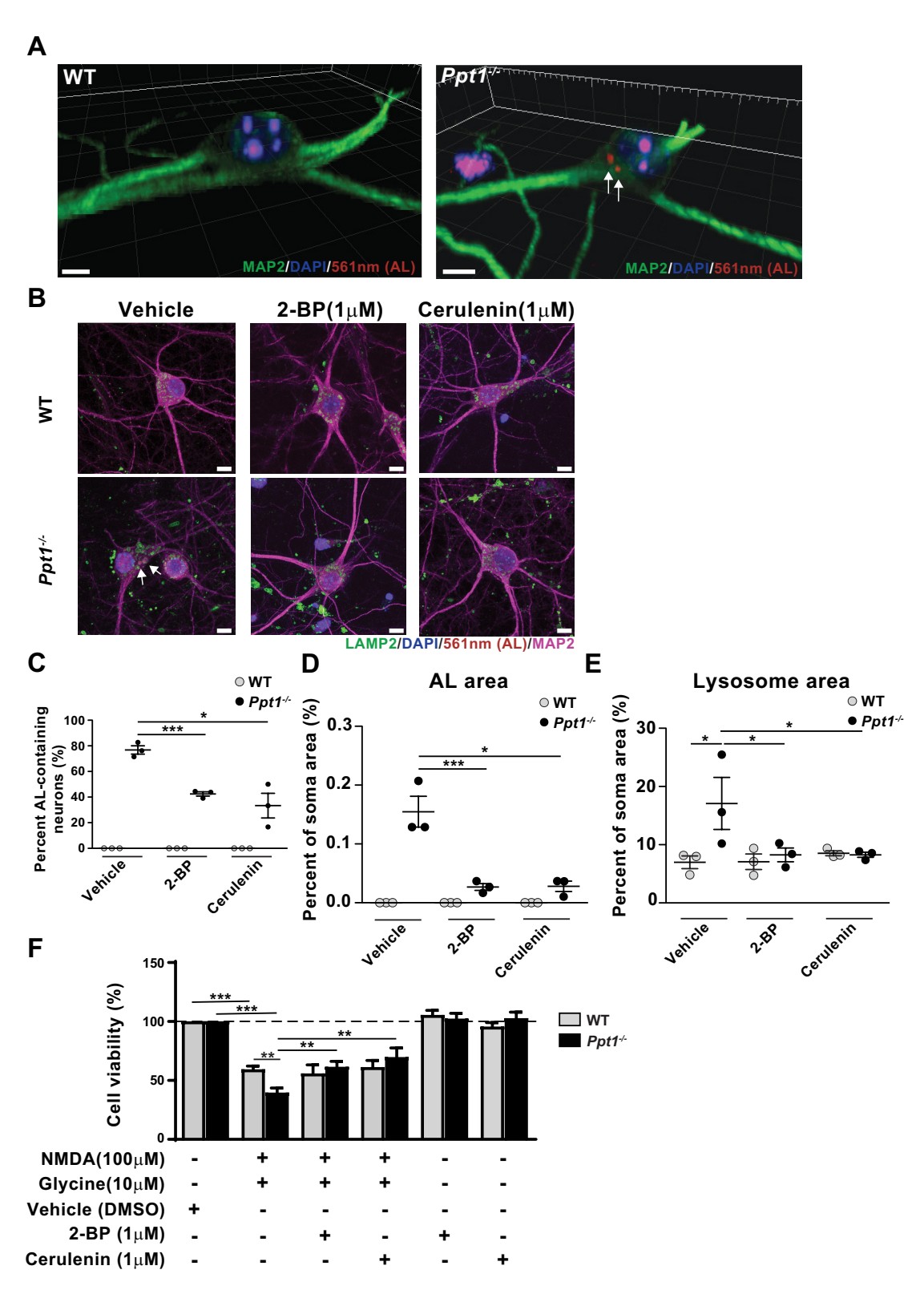

**Figure 9.** Palmitoylation inhibitors rescue enhanced vulnerability to NMDA-mediated excitotoxicity in *Ppt1⁻/⁻* cultured neurons. (A) 3D reconstructions of a WT and *Ppt1⁻/⁻* neuron at DIV20. Arrows point to AL deposits. Scale bar = 5 µm (B) Representative collapsed z-stacks of WT and *Ppt1⁻/⁻* DIV20 neurons, demonstrating accumulations of ALs (arrows) within the soma, particularly within LAMP2-positive vesicles, of *Ppt1⁻/⁻* neurons. Note the enlarged lysosomes in *Ppt1⁻/⁻*, vehicle-treated neurons (see *Figure 9—figure supplement 1*). Scale bar = 10 µm (C) Quantification of the percentage of

*Figure 9 continued*

AL-containing neurons at DIV20 with or without the palmitoylation inhibitors, 2-bromopalmitate (2 BP, 1 µM) and cerulenin (1 µM), treatment for 6d. (**D**) The percentage of soma area occupied by ALs with or without the palmitoylation inhibitors, 2 BP (1 µM) and cerulenin (1 µM), treatment for 6d. (**E**) Quantification of the percentage of soma area occupied by lysosomes (LAMP-2-positive vesicles) with and without palmitoylation inhibitor, 2 BP (1 µM) and cerulenin (1 µM), treatment for 6 days. WT, $Ppt1^{-/-}$, and drug treatment conditions were compared (7–10 neurons/group/experiment, n = 3 independent experiments) by two-way ANOVA followed by Tukey's post-hoc test and significance indicated as follows: *p<0.05, ***p<0.001 where indicated. (**F**) Quantification of cellular viability in DIV18-20 WT and $Ppt1^{-/-}$ neurons treated with NMDA and glycine (100/10 µM) with or without pretreatment with vehicle (DMSO) only, 2 BP (1 µM) or cerulenin (1 µM). Values for treatment with 2 BP (1 µM) or cerulenin (1 µM) in the absence of NMDA and glycine are also shown. WT, $Ppt1^{-/-}$, and drug treatment conditions were compared (n = 4 independent experiments, in duplicate) by two-way ANOVA followed by Tukey's post-hoc test and significance indicated as follows: **p<0.01, ***p<0.001 where indicated. Error bars represent s.e.m.
DOI: https://doi.org/10.7554/eLife.40316.015

The following figure supplement is available for figure 9:

**Figure supplement 1.** ALs accumulate in enlarged lysosomes of Ppt1-/- neurons.
DOI: https://doi.org/10.7554/eLife.40316.016

(*Figure 11B*). Remarkably, the palmitoylation states of PSD-95 were comparable between $Ppt1^{-/-}$ and WT neurons (*Figure 11C*), suggesting that PSD-95 may not be a PPT1 substrate. This finding is consistent with previous results showing that PSD-95 is not depalmitoylated by PPT1 (*Yokoi et al., 2016*). Further, in line with previous data (*El-Husseini et al., 2002*; *Fukata et al., 2013*), 2 BP treatment decreased the relative palmitoylation level (ratio of palm/non-palm) of PSD-95 by nearly 50% in WT neurons (*Figure 11C*). However, we did not observe this effect in $Ppt1^{-/-}$ neurons. Also, cerulenin treatment had no consistent effects on PSD-95 levels or palmitoylation state in WT or $Ppt1^{-/-}$ cells (*Figure 11B and C*). However, the specificity of palmitoylation inhibitors is incompletely understood and compensatory mechanisms may restore PSD-95 palmitoylation due to chronic low-dose inhibitor treatment.

We performed the same analysis on another well-studied palmitoylated protein, Fyn kinase (*Figure 11D*). Fyn is a prominent member of the Src family kinases that phosphorylates and thereby stabilizes GluN2B at the synaptic surface (*Prybylowski et al., 2005*; *Trepanier et al., 2012*). Further, Fyn palmitoylation is important for its localization to the plasma membrane, where it may interact with GluN2B (*Sato et al., 2009*). Hence, Fyn hyperpalmitoylation can be a mechanism by which GluN2B retention may be enhanced in $Ppt1^{-/-}$ neurons. Indeed, total levels of Fyn were increased in $Ppt1^{-/-}$ neurons compared to WT, and were significantly suppressed by 2 BP and cerulenin treatment in both WT and $Ppt1^{-/-}$ neurons (*Figure 11E*). 2 BP and cerulenin treatments also significantly reduced the ratio of palmitoylated/non-palmitoylated Fyn in WT and $Ppt1^{-/-}$ neurons (*Figure 11F*). These findings imply that palmitoylation of Fyn regulates its stability and that Fyn hyperpalmitoylation may play a vital role in the stagnation of GluN2B to GluN2A subunit switch (*Figure 11E and F*).

Next, we examined the palmitoylation state of GluN2B (*Figure 11G*). First, total GluN2B levels were comparable between WT and $Ppt1^{-/-}$, vehicle-treated neurons at DIV18 (*Figure 11G and H*). While 2 BP had no effect on total GluN2B levels in WT neurons, 2 BP treatment decreased total GluN2B in $Ppt1^{-/-}$ neurons compared to vehicle-treated cells (*Figure 11H*). Cerulenin had the same effect (*Figure 11H*). Importantly, both 2 BP and cerulenin corrected the ratio of palmitoylated/non-palmitoylated GluN2B in $Ppt1^{-/-}$ neurons, with values approaching those of vehicle-treated WT cells (*Figure 11I*). No effect was observed in WT neurons treated with cerulenin. The latter results indicate that GluN2B palmitoylation state is less sensitive to chronic, low-dose palmitoylation inhibitor treatment than Fyn palmitoylation state. One possibility, therefore, is that enhanced

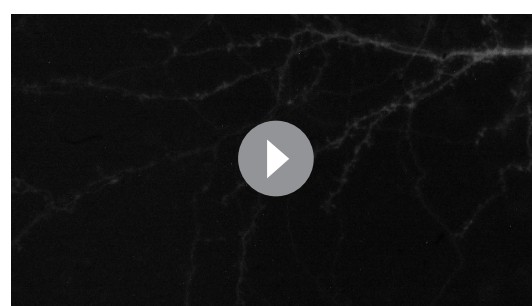

**Video 5.** Spontaneous calcium activity in DIV18 $Ppt1^{-/-}$ neuron treated with 2 BP (1 µM, from DIV12-18). Representative video of spontaneous neuronal calcium activity in a $Ppt1^{-/-}$ cultured neuron that was treated from DIV12-18 with 2 BP, at a dose of 1 µM. The last treatment was 4–6 hr before the imaging session.
DOI: https://doi.org/10.7554/eLife.40316.018

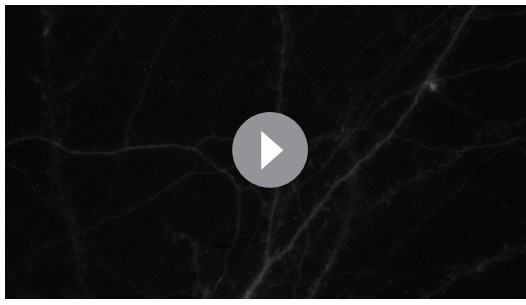 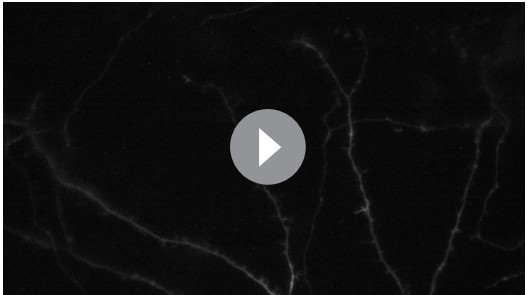

**Video 6.** Spontaneous calcium activity in DIV18 *Ppt1⁻/⁻* neuron treated with cerulenin (1 µM, from DIV12-18). Representative video of spontaneous neuronal calcium activity in a *Ppt1⁻/⁻* cultured neuron that was treated from DIV12-18 with cerulenin, at a dose of 1 µM. The last treatment was 4–6 hr before the imaging session. DOI: https://doi.org/10.7554/eLife.40316.019

**Video 7.** Spontaneous calcium activity in DIV18 *Ppt1⁻/⁻* neuron treated with vehicle (DMSO, from DIV12-18). Representative video of spontaneous neuronal calcium activity in a *Ppt1⁻/⁻* cultured neuron that was treated from DIV12-18 with DMSO as vehicle. The last treatment was 4–6 hr before the imaging session. DOI: https://doi.org/10.7554/eLife.40316.020

surface retention of GluN2B-containing NMDARs in *Ppt1⁻/⁻* neurons results from Fyn hyperpalmitoylation. Alternatively, hyperpalmitoylation of GluN2B may directly lead to enhanced surface retention of GluN2B-containing NMDARs in *Ppt1⁻/⁻* neurons (*Mattison et al., 2012*). Nevertheless, these data are in line with *Figure 10C*, which shows the frequency of calcium influxes in *Ppt1⁻/⁻* neurons is robustly decreased by treatment with 2 BP or cerulenin.

Finally, we examined the palmitoylation state of GluN2A at baseline and in response to palmitoylation inhibitor treatment (*Figure 11J*). As we found in our biochemical analyses of the cortical homogenates (*Figures 2* and *5*), *Ppt1⁻/⁻* neurons showed decreases in the band intensity of total GluN2A level as compared to WT (*Figure 11J and K*). The palmitoylation state (ratio of palm/non-palm) of GluN2A, however, was unchanged between WT and *Ppt1⁻/⁻* at baseline (*Figure 11L*), suggesting that PPT1 is not directly involved in the palmitoylation state of GluN2A. Interestingly, chronic treatment with 2 BP and cerulenin had dissimilar effects on GluN2A levels and palmitoylation state in WT and *Ppt1⁻/⁻* neurons. In WT cells, 2 BP had no effect on GluN2A levels or palmitoylation state (*Figure 11K and L*), indicating that the chronic low-dose treatment does not intervene the GluN2A depalmitoylation in WT neurons. In contrast, 2 BP treatment in *Ppt1⁻/⁻* neurons robustly increased both the total level and palmitoylation state of GluN2A, resulting in the nearly equal representation of two distinct GluN2A palmitoylated species (*Figure 11K and L*). Differing from 2 BP, cerulenin treatment modestly decreased total GluN2A levels and GluN2A palmitoylation state in WT cells (*Figure 11K and L*). In contrast, cerulenin treatment of *Ppt1⁻/⁻* cells increased the total GluN2A protein level (*Figure 11K and L*), albeit not nearly as robustly as 2 BP. Overall, these results illustrate complex and potentially indirect effects of palmitoylation inhibitor treatment on the palmitoylation state of GluN2A in *Ppt1⁻/⁻* neurons. While there may be other possibilities, the data suggest that 2 BP treatment may have corrected a defect in palmitoylation in *Ppt1⁻/⁻* neurons (e.g. Fyn hyperpalmitoylation), thereby initiating or facilitating the GluN2B to GluN2A switch. Nevertheless, the increase in GluN2A levels and palmitoylation in *Ppt1⁻/⁻* cells may ultimately account for the beneficial effects of inhibitor treatment in our complementary analyses (*Figures 9* and *10*).

In aggregate, these data indicate hyperpalmitoylation of Fyn and GluN2B, and point to mechanisms by which chronic low-dose palmitoylation inhibitor treatment may decrease the synaptic stabilization of GluN2B, thereby reducing calcium load in *Ppt1⁻/⁻* neurons and mitigating the enhanced susceptibility to excitotoxicity. Further, these data imply that the progression of CLN1 may be mediated by the palmitoylation of Fyn kinase, which is also being targeted for the treatment of Alzheimer's disease (*Kaufman et al., 2015*; *Nygaard et al., 2015*).

## Discussion

Since the development of the first *Ppt1⁻/⁻* mouse and knock-in mouse model of CLN1, much progress has been made in understanding the temporal, regional, and cell-type specific effects of lipofuscin

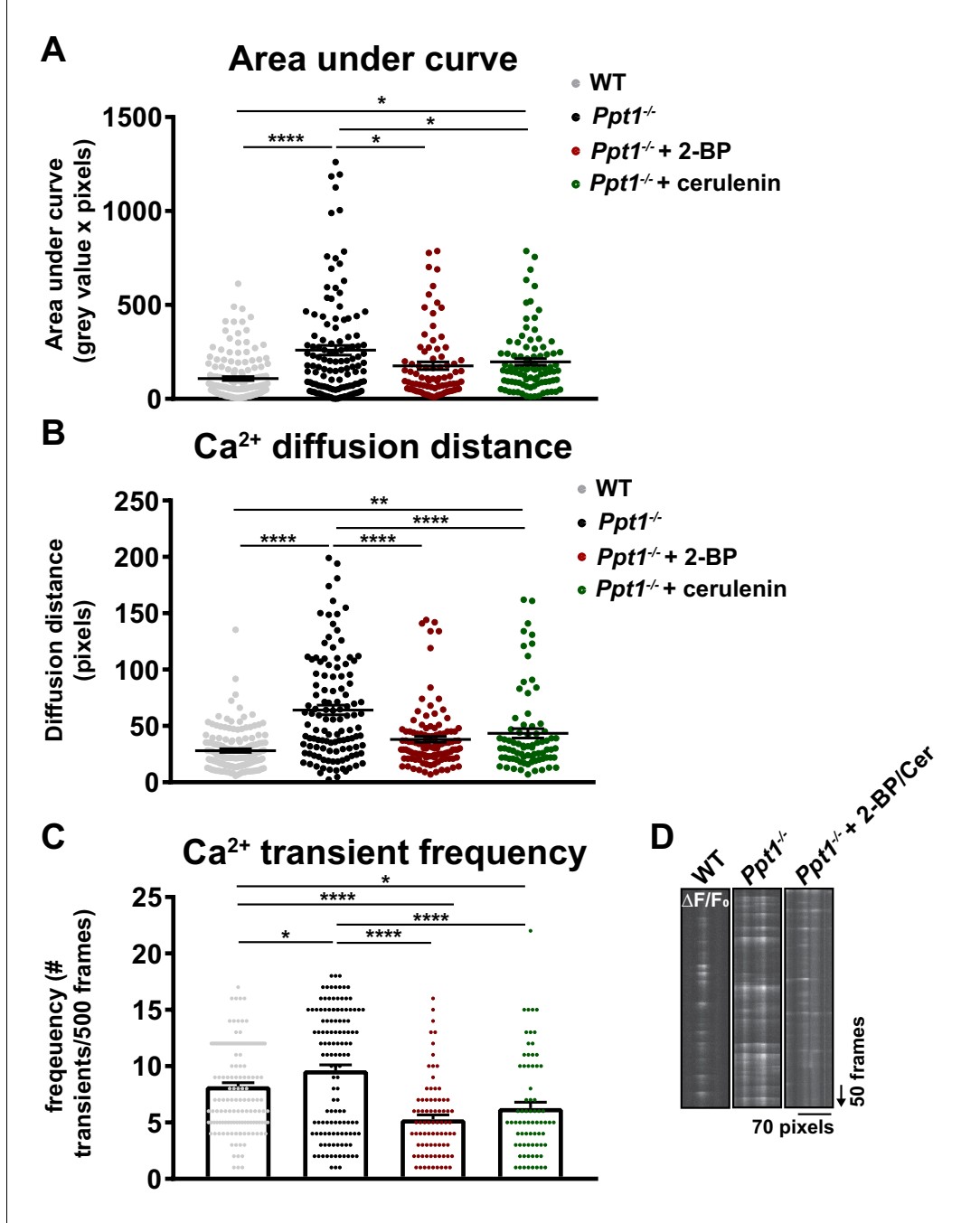

**Figure 10.** Palmitoylation inhibitor treatment partially reverses pathological calcium dynamics in *Ppt1*$^{-/-}$ neurons. A subset of *Ppt1*$^{-/-}$ cells treated with the palmitoylation inhibitors, 2 BP (1 µM) or cerulenin (1 µM) from DIV12-18 were transfected with GCaMP3 and imaged in the absence of Mg$^{2+}$ for 5 min. Sites of calcium influx (spontaneous synaptic activity, ΔF/F$_0$) were analyzed. These data were compared to WT and *Ppt1*$^{-/-}$ groups from *Figure 7*. (A) Quantification of calcium transient area under the curve (AUC) in WT, *Ppt1*$^{-/-}$, *Ppt1*$^{-/-}$ + 2 BP, and *Ppt1*$^{-/-}$ + cerulenin-treated groups. (B) Quantification of calcium transient diffusion distance from WT, *Ppt1*$^{-/-}$, *Ppt1*$^{-/-}$ + 2 BP, and *Ppt1*$^{-/-}$ + cerulenin treated groups. (C) Quantification of calcium transient frequency (# transients/500 frames) for each synaptic site between WT, *Ppt1*$^{-/-}$, *Ppt1*$^{-/-}$ + 2 BP, and *Ppt1*$^{-/-}$ + cerulenin-treated groups. (D) Representative kymographs displaying the calcium influx (ΔF/F$_0$) at one synaptic site (pixels, X-axis) over the course of 500 frames (arrow represents direction of time in frames) from WT (left), *Ppt1*$^{-/-}$ (middle), and *Ppt1*$^{-/-}$ + palmitoylation inhibitor treatment groups. For experiments in *Figure 10A–C*, WT, *Ppt1*$^{-/-}$, *Ppt1*$^{-/-}$ + 2 BP, and *Ppt1*$^{-/-}$ + cerulenin groups were compared (n = 185 synaptic sites (WT), n = 131 synaptic sites (*Ppt1*$^{-/-}$), n = 85 synaptic sites (*Ppt1*$^{-/-}$ + 2 BP), n = 82 synaptic sites (*Ppt1*$^{-/-}$ + cerulenin); 3–4 neurons/group; three individual experiments) by one-way ANOVA followed by Tukey's post-hoc test and the significance was indicated as follows: *p<0.05,**p<0.01 ****p<0.0001 where indicated. Dots represent values for individual synaptic sites. Error bars represent s.e.m.

DOI: https://doi.org/10.7554/eLife.40316.017

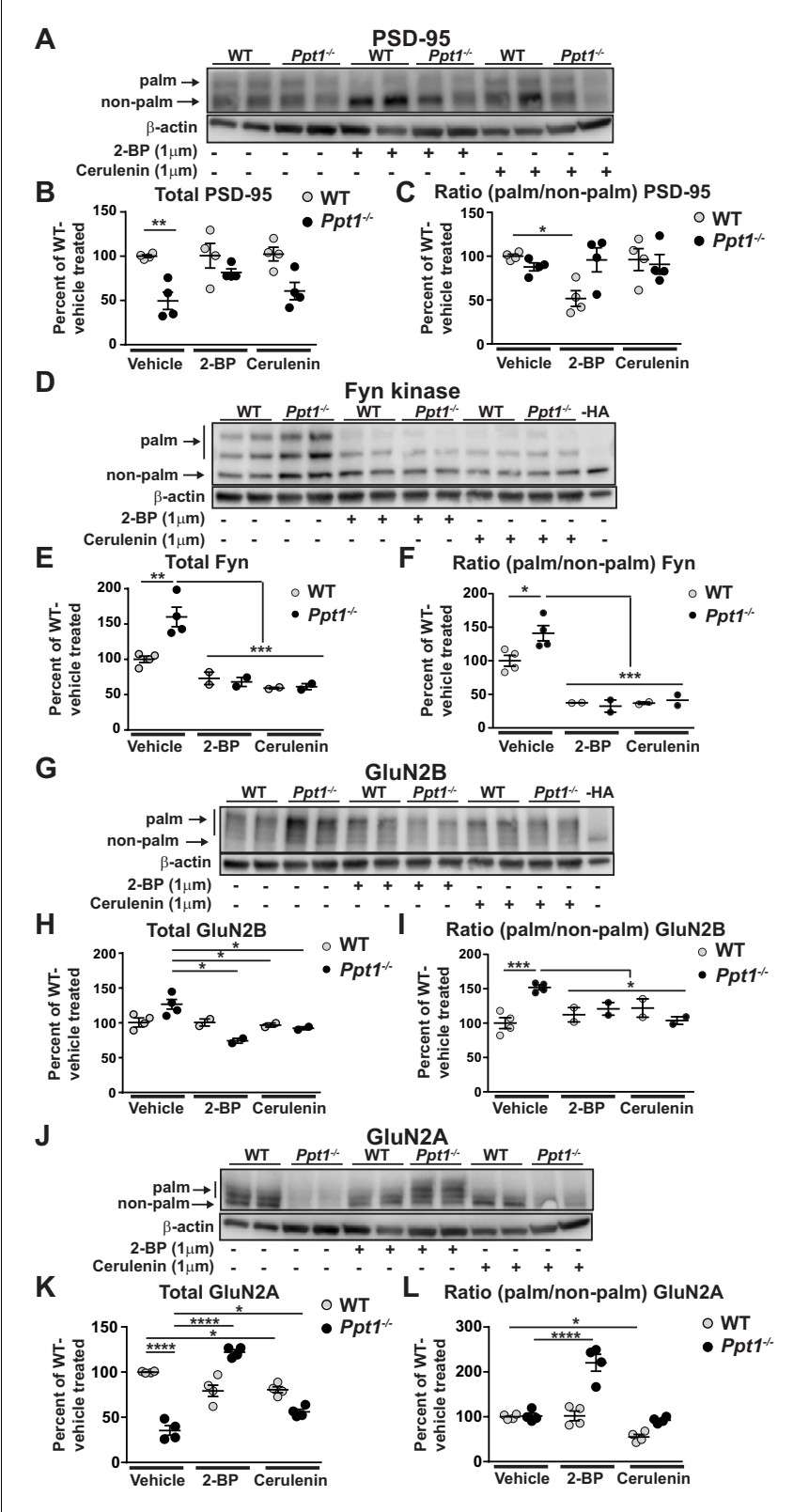

**Figure 11.** Hyperpalmitoylation of Fyn kinase and GluN2B is reversed in *Ppt1*[-/-] primary cortical neurons by palmitoylation inhibitor treatment. (**A**) Representative post-APEGS immunoblot of PSD-95 with β-actin loading control. (**B**) Quantification of total PSD-95 levels following chronic (7d) treatment with vehicle or the palmitoylation inhibitors, 2 BP (1 μM) or cerulenin (1 μM) where indicated. (**C**) Quantification of the ratio of palmitoylated/non-palmitoylated PSD-95 levels following chronic (7d) treatment with vehicle or the palmitoylation inhibitors, 2 BP (1 μM) or cerulenin (1 μM) where

*Figure 11 continued on next page*

*Figure 11 continued*

indicated. (D) Representative post-APEGS immunoblot of Fyn kinase with β-actin loading control and minus hydroxylamine (-HA) control. (E) Quantification of total Fyn kinase levels following chronic (7d) treatment with vehicle or the palmitoylation inhibitors, 2 BP (1 µM) or cerulenin (1 µM) where indicated. (F) Quantification of the ratio of palmitoylated/non-palmitoylated Fyn kinase levels following chronic (7 days) treatment with vehicle or the palmitoylation inhibitors, 2 BP (1 µM) or cerulenin (1 µM) where indicated. (G) Representative post-APEGS immunoblot of GluN2B with β-actin loading control and minus hydroxylamine (-HA) control. (H) Quantification of total GluN2B levels following chronic (7d) treatment with vehicle or the palmitoylation inhibitors, 2 BP (1 µM) or cerulenin (1 µM) where indicated. (I) Quantification of the ratio of palmitoylated/non-palmitoylated GluN2B levels following chronic (7 days) treatment with vehicle or the palmitoylation inhibitors, 2 BP (1 µM) or cerulenin (1 µM) where indicated. (J) Representative post-APEGS immunoblot of GluN2A with β-actin loading control. (K) Quantification of total GluN2A levels following chronic (7d) treatment with vehicle or the palmitoylation inhibitors, 2 BP (1 µM) or cerulenin (1 µM) where indicated. (L) Quantification of the ratio of palmitoylated/non-palmitoylated GluN2A levels following chronic (7d) treatment with vehicle or the palmitoylation inhibitors, 2 BP (1 µM) or cerulenin (1 µM) where indicated. DMSO was used for vehicle treatment. For experiments in *Figure 11A and D*, WT, *Ppt1*$^{-/-}$, WT + 2 BP, WT + cerulenin, *Ppt1*$^{-/-}$ + 2 BP, and *Ppt1*$^{-/-}$ + cerulenin treatment groups were compared (n = 4 independent experiments) by two-way ANOVA followed by Tukey's post-hoc test and significance indicated as follows: *p<0.05, **p<0.01, ****p<0.0001 where indicated. For experiments in *Figure 11B and C*, WT, *Ppt1*$^{-/-}$, WT + 2 BP, WT + cerulenin, *Ppt1*$^{-/-}$ + 2 BP, and *Ppt1*$^{-/-}$ + cerulenin were compared (n = 2–4 independent experiments) at each time point using two-way ANOVA followed by Tukey's post-hoc test and the significance was indicated as follows: *p<0.05, **p<0.01, ***p<0.001 where indicated. Error bars represent s.e.m.

DOI: https://doi.org/10.7554/eLife.40316.021

accumulation and neuronal degeneration, particularly in late stage disease (*Gupta et al., 2001*; *Bible et al., 2004*; *Kielar et al., 2007*; *Bouchelion et al., 2014*). In addition, comprehensive data characterizing the behavioral dysfunction of the *Ppt1*$^{-/-}$ mouse has recapitulated clinical symptoms of the disease (*Dearborn et al., 2015*). Recent data demonstrate that PPT1 localizes to synaptic compartments and influences presynaptic localization and mobility of prominent presynaptic proteins, including SNAP25 (*Kim et al., 2008*). These findings correlate with histological and electrophysiological findings in cultured *Ppt1*$^{-/-}$ neurons, demonstrating a depletion of presynaptic vesicle pool size (*Virmani et al., 2005*). Moreover, presynaptic protein localization and function are altered in CLN1 models and in human tissue (*Kanaani et al., 2004*; *Kim et al., 2008*; *Aby et al., 2013*).

In the current study, we have identified a role for PPT1 in postsynaptic maturation in the *Ppt1*$^{-/-}$ mouse model of CLN1. Our principal finding demonstrates a role for PPT1 in the regulation of NMDAR composition and function that leaves *Ppt1*$^{-/-}$ neurons vulnerable to excitotoxic insult, which is alleviated by chronic low-dose palmitoylation inhibitor treatment. Together, these data implicate dysregulated GluN2 subunit switch as a major pathogenic mechanism in CLN1.

## Stagnation of GluN2B to GluN2A subunit switch in the *Ppt1*$^{-/-}$ mouse visual cortex

During early postnatal development, NMDARs switch their subunit composition from primarily containing GluN2B subunits to predominantly containing GluN2A subunits (*Carmignoto and Vicini, 1992*; *Sheng et al., 1994*; *Stocca and Vicini, 1998*). The increased contribution of GluN2A subunits is accompanied by several distinctive changes in NMDAR-mediated synaptic currents, including a shortening of the decay time (*Carmignoto and Vicini, 1992*). In the rodent visual cortex, developmental decrease of the NMDAR-EPSC decay time and the corresponding switch from GluN2B- to GluN2A-diheteromeric receptors occurs between the second and fourth postnatal weeks (*Carmignoto and Vicini, 1992*; *Quinlan et al., 1999a*; *Quinlan et al., 1999b*; *Philpot et al., 2001*). The GluN2 subunit switch is not absolute, and GluN2B subunits remains expressed in many regions of the adult brain (*Lopez de Armentia and Sah, 2003*). Consequently, synaptic NMDARs are predominantly diheteromeric GluN1/GluN2A and triheteromeric GluN1/GluN2A/GluN2B receptors (*Luo et al., 1997*; *Tovar and Westbrook, 1999*; *Tovar et al., 2013*), while extrasynaptic sites are enriched in GluN2B-containing receptors (*Carmignoto and Vicini, 1992*; *Rumbaugh and Vicini, 1999*; *Tovar and Westbrook, 1999*).

Our biochemical analyses show reductions in the protein amount of GluN2A subunit and their preferential synaptic scaffold, PSD-95, in the *Ppt1*$^{-/-}$ mouse visual cortex at distinct developmental time points (P33-P60). These alterations correlate with the prolongation of the evoked NMDAR-EPSC decay time in *Ppt1*$^{-/-}$ layer II/III cortical neurons. One may anticipate the prolonged decay time in *Ppt1*$^{-/-}$ neurons is more sensitive to a GluN2B-specific inhibitor. However, both WT and *Ppt1*$^{-/-}$

neurons equally responded to Ro 25–6981 (*Figure 3E*). Importantly, triheteromeric NMDARs have intermediate decay kinetics (between purely GluN2A- or GluN2B-containing diheteromeric receptors) and are relatively insensitive to GluN2B-specific inhibitors (*Stroebel et al., 2018*). Still, the GluN2B subunit may dominate specific features of these triheteromeric NMDARs such as their recycling rate or activation during synaptic plasticity (*Tang et al., 2010*; *Delaney et al., 2013*). We demonstrate that GluN2B is hyperpalmitoylated in *Ppt1$^{-/-}$* neurons (*Figure 11C*). Therefore, the dysregulated GluN2 subunit switch in *Ppt1$^{-/-}$* neurons may be explained by the growing presence of the triheteromeric receptors at P42 (and anticipate the same results between P33 and P60) in the cortex (*Luo et al., 1997*). Specifically, we propose that increased palmitoylation of GluN2B subunits leads to more stable assembly of triheteromeric receptors and their accumulation at postsynaptic sites.

However, the presence of triheteromeric receptors does not fully explain the Ca$^{2+}$ imaging data demonstrating that Ro 25–6981 treatment sufficiently inhibits extrasynaptic Ca$^{2+}$ influx in vitro (*Figure 7*), which is likely mediated by diheteromeric GluN2B-NMDARs. In rat hippocampal neurons, although the ifenprodil-sensitive component of synaptic and extrasynaptic NMDAR populations declines with maturation, the majority of extrasynaptic NMDARs remain sensitive to ifenprodil (diheteromeric GluN2B-NMDARs) even into synaptic maturity (DIV13-19)(*Thomas et al., 2006*). This sustained extrasynaptic population of diheteromeric GluN2B-NMDARs would be represented in our in vitro imaging experiments (in Mg$^{2+}$-free solution) and be blocked by bath application of Ro 25–6981, constraining the remaining activity to synaptic NMDARs that more closely resemble WT neurons (*Figure 7*). Another explanation is that immature spine structure may have altered the dendritic distribution of GluN2B-containing diheteromeric receptors in *Ppt1$^{-/-}$* neurons, limiting the representation of these Ro 25–6981-sensitive NMDARs evoked by single pulse of synaptic activation as in *Figure 3E*. These differences in experimental preparation (dissociated neurons vs. slice and electrophysiology vs. Ca$^{2+}$ imaging) may therefore contribute to the observed variation in Ro 25–6981 efficacy in vivo and in vitro.

## Candidate PPT1 substrates that regulate the GluN2B to GluN2A subunit switch

Initially, we predicted synaptic markers, particularly PSD-95, would be hyperpalmitoylated and over-represented at postsynaptic sites, since their synaptic distribution depends on the balance between palmitoylation and depalmitoylation (*Craven et al., 1999*; *El-Husseini et al., 2000a*; *Jeyifous et al., 2016*). As PSD-95 facilitates the GluN2 subunit switch and preferentially interacts with GluN2A, we also hypothesized an increase in the GluN2A subunit. However, our biochemical data indicate reductions in the total amount as well as the synaptic incorporation of GluN2A and PSD-95 in the PPT1-deficient brain (*Figure 2*, *Figure 2—figure supplement 1*). Interestingly, a recent study suggests that PSD-95 is depalmitoylated by the α/β hydrolase domain-containing 17 (ABHD17) family of depalmitoylating enzymes, not PPT1 (*Yokoi et al., 2016*). Our results support this notion, as PSD-95 palmitoylation state is comparable between WT and *Ppt1$^{-/-}$* neurons (*Figure 11B*). These findings illustrate that depalmitoylases demonstrate substrate specificity which may be important for the regulation of coordinated, long-term changes at the synapse.

Our biochemical analyses demonstrate that the GluN2B to GluN2A switch is disrupted both in vivo (*Figure 2A–D*) and in vitro (*Figure 5A–D*) in *Ppt1$^{-/-}$* neurons and that PPT1 expression in the WT visual cortex correlates tightly with the timing of this disruption (*Figure 2E*). Further, both electrophysiological (*Figure 3*) and calcium imaging data (*Figure 7*) indicate that *Ppt1$^{-/-}$* neurons have more GluN2B-predominant extrasynaptic NMDARs. Finally, we demonstrate that GluN2B is hyperpalmitoylated in Ppt1$^{-/-}$ in vitro (*Figure 11C*). These findings raise the possibility that GluN2B is a PPT1 substrate and thus, lack of PPT1-mediated GluN2B depalmitoylation partly drives disease symptoms. Indeed, protein palmitoylation generally enhances the half-life of proteins (*Linder and Deschenes, 2007*) and, in the case of GluN2B, specifically enhances its phosphorylation and consequent surface retention (*Hayashi et al., 2009*). Thus, GluN2B hyperpalmitoylation alone may alter NMDAR function enough to drive some of the functional deficits described in *Ppt1$^{-/-}$* neurons herein.

Alternatively, the lack of PPT1 function may indirectly set the stage for an overrepresentation of GluN2B at excitatory synapses and consequent pathogenic activity. A primary candidate for such an indirect mechanism involves Src family kinase Fyn, which regulates GluN2B protein conformation, surface retention, and fine-scale synaptic localization (*Prybylowski et al., 2005*; *Nakazawa et al.,*

*2006*; *Mattison et al., 2012*). Fyn is palmitoylated and its subcellular localization is palmitoylation-dependent (*Koegl et al., 1994*). Further, Fyn kinase is developmentally regulated (*Umemori et al., 1992*; *Inomata et al., 1994*) and a major palmitoylated downstream kinase of reelin signaling (see below) that phosphorylates GluN2B, affecting its surface stabilization (*Alland et al., 1994*; *Koegl et al., 1994*; *Prybylowski et al., 2005*; *Kang et al., 2008*). Crucially, palmitoylation of GluN2B enhances Fyn-mediated phosphorylation at Tyr1472 and thereby inhibits its internalization (*Hayashi et al., 2009*). Thus, Fyn hyperactivation may be responsible for the dysregulation of GluN2B to GluN2A switch in *Ppt1⁻ᐟ⁻* neurons. Specifically, hyperpalmitoylated Fyn kinase may lead to enhanced phosphorylation and surface retention of GluN2B-containing NMDARs, limiting access to alternative depalmitoylating enzymes which may act in recycling endosomes or other cellular compartments, resulting in GluN2B hyperpalmitoylation. Indeed, the palmitoylation state of Fyn is more sensitive to palmitoylation inhibitors (*Figure 11B*) than GluN2B (*Figure 11C*). Notably, 2BP-treated *Ppt1⁻ᐟ⁻* neurons exhibit an increase in GluN2A palmitoylation state (*Figure 11D*). While the reason for change is not entirely clear, one possibility is that the palmitoylation state of Fyn is more sensitive to the chronic low-dose inhibition than that of GluN2 subunits. In this scenario, suppression of Fyn palmitoylation attenuates its function, thereby exaggerating the GluN2 subunit switch in *Ppt1⁻ᐟ⁻* neurons.

There are several other mechanisms underlying this GluN2 subunit switch that may be affected by lack of PPT1. These signaling pathways include reelin, Wnt-5a, and mGluR5 (*Groc et al., 2007*; *Cerpa et al., 2011*; *Matta et al., 2011*). The accumulation of reelin at excitatory synapses during development, for example, mobilizes GluN2B-containing NMDARs and enhances the synaptic contribution of GluN2A-containing NMDARs (*Groc et al., 2007*; *Iafrati et al., 2014*). Similarly, evoked activation of mGluR5 at hippocampal synapses is necessary for incorporation of GluN2A-containing NMDARs, and mGluR5-null mice demonstrate deficient GluN2B to GluN2A switching (*Matta et al., 2011*). Importantly, Wnt-5a, mGluR5, and Fyn kinase (downstream of reelin signaling) are directly regulated by palmitoylation state (*Kurayoshi et al., 2007*; *Yokoi et al., 2016*), suggesting that disruptions in protein depalmitoylation may lead to impaired synaptic maturation through several pathways. Further study is needed to elucidate precisely how PPT1 influences the GluN2B to GluN2A switch and if Fyn is indeed a key mediator. Nevertheless, we have shown that the lack of functional PPT1 results in aberrant surface retention of GluN2B-containing NMDAR complexes, either directly or indirectly, thereby impeding the developmental switch to GluN2A-containing receptors.

## Excitotoxicity and NMDAR regulation

Patients afflicted with later-onset NCLs typically exhibit an enlarged VEP prior to degeneration, concurrent with seizure (*Pampiglione and Harden, 1977*; *Haltia, 2006*; *Pagon et al., 2013*). This phenomenon has not been directly observed in CLN1, though this may be due to the rapid degeneration and advanced pathology at time of diagnosis for these patients. Nevertheless, it is conceivable that disrupted GluN2 subunit switch contributes to hyperexcitability in CLN1 and thereby accelerates cell death, leading to the rapid degeneration of neuronal circuits. Indeed, recent evidence link GluN2 subunit composition and NMDAR localization to opposing downstream transcriptional programs (*Martel et al., 2009*; *Martel et al., 2012*; *Hardingham and Bading, 2010*). Specifically, GluN2A-containing NMDARs in the postsynaptic density activate cyclic-AMP response element binding protein (CREB) and other transcription factors associated with cell-survival and learning. In contrast, extrasynaptic, GluN2B-containing NMDARs preferentially trigger pro-apoptotic signaling pathways and cause inhibition of CREB (*Hardingham and Bading, 2002*; *Hardingham et al., 2002*). Although this system is likely more intricate than described here (*Thomas et al., 2006*), these previous studies are consistent with our observations that *Ppt1⁻ᐟ⁻* neurons are biased toward extrasynaptic calcium transients (*Figure 7* and *Video 2*) and that they are more susceptible to excitotoxicity (*Figures 8* and *9*). These data are also in agreement with previous studies demonstrating markedly enhanced NMDA-mediated toxicity in *Ppt1⁻ᐟ⁻* neurons and improved behavioral phenotype of *Ppt1⁻ᐟ⁻* mice treated with the NMDAR antagonist, memantine (*Finn et al., 2013*). Furthermore, the most significant outcome of this study is that palmitoylation inhibitors mitigated the pro-apoptotic predisposition of *Ppt1⁻ᐟ⁻* neurons in vitro (*Figure 9*).

The incorporation of GluN2A into NMDARs is experience-dependent (*Quinlan et al., 1999b*; *Quinlan et al., 1999a*). Therefore, an intriguing possibility is that *Ppt1⁻ᐟ⁻* neurons in sensory cortices are unable to tolerate normal sensory experiences, in part because this experience-dependent

GluN2 subunit switch is disrupted. Indeed, PPT1-defeciency results in selective degeneration of thalamic nuclei and primary sensory cortices (*Bible et al., 2004*; *Kielar et al., 2007*). Further, PPT1 expression, as we demonstrate herein (*Figure 2E*, *Figure 2—figure supplement 1E*), is developmentally regulated in WT rodents and may mediate this switching phenomenon (*Suopanki et al., 1999a*; *Suopanki et al., 1999b*). Together, we argue that intact PPT1 plays a critical role in regulating NMDAR functional properties in response to external stimuli, thereby facilitating synaptic maturation and preventing excitotoxicity. Whether manipulating neuronal activity or experience-dependent synaptic plasticity ameliorates disease progression remains unknown and is a focus of ongoing experiments.

## Dendritic spine immaturity induced by lack of PPT1

During neurodevelopment, rapid spinogenesis involves filopodial formation followed by molecular and structural changes that lead to dendritic spine maturation in the adult brain. In the current study, we demonstrate that dendritic spines in $Ppt1^{-/-}$ neurons are longer, thinner, and show increased density as compared to WT (*Figures 4* and *6*), which are generally indications of spine immaturity. Indeed, the processes underlying spinogenesis are regulated by various palmitoylated proteins including the ones we have examined in this study. For instance, it is established that Src family kinase activity, including Fyn, mediates biochemical changes that lead to filopodia and dendritic spine formation in neurons (*Morita et al., 2006*; *Webb et al., 2007*; *Babus et al., 2011*; *Formoso et al., 2015*). GluN2B activity also enhances filopodial formation in hippocampal neurons, and application of GluN2B-specific blockers inhibits this effect (*Henle et al., 2012*). Further, GluN2B hyperpalmitoylation at specific sites stabilizes GluN2B-containing receptors at the cell surface (*Mattison et al., 2012*). Thus, it is plausible that hyperpalmitoylation of GluN2B in $Ppt1^{-/-}$ cells increases filopodial formation through enhanced expression of surface NMDARs, resulting in an increased spine density (*Figures 4* and *6*). Whether hyperpalmitoylation of GluN2B and Fyn are directly responsible for the altered spine morphology in $Ppt1^{-/-}$ neurons awaits further studies.

Other palmitoylated proteins may also account for aberrant spine formation in $Ppt1^{-/-}$ neurons. Increased spine density in $Ppt1^{-/-}$ neurons suggests that filopodial organizer proteins may be substrates of PPT1 or are indirectly affected by the enzyme. Several neuronal proteins readily induce filopodial formation in a palmitoylation-dependent manner in cultured neurons (*Patterson and Skene, 1994*; *Gauthier-Campbell et al., 2004*). For example, palmitoylation of the PPT1 substrate growth-associated protein 43 (GAP43) regulates filopodial formation (*Kutzleb et al., 1998*; *Gauthier-Campbell et al., 2004*; *Arstikaitis et al., 2008*), and its protein amount is increased in the $Ppt1^{-/-}$ brain beginning at 1 month (*Zhang et al., 2006*). Cdc42 is another palmitoylated protein that may accelerate neurite formation in $Ppt1^{-/-}$ neurons (*Gauthier-Campbell et al., 2004*; *Kang et al., 2008*). Hyperpalmitoylation of these proteins in the absence of PPT1 may accelerate their activity and lead to excessive filopodial formation in $Ppt1^{-/-}$ neurons (*Patterson and Skene, 1999*).

Filopodial formation is generally followed by spine maturation, which is facilitated by the palmitoylation and localization of PSD-95 at the postsynaptic membrane (*Craven et al., 1999*; *El-Husseini et al., 2000a*; *Yoshii et al., 2011*; *Jeyifous et al., 2016*). Synaptosomes derived from $Ppt1^{-/-}$ mouse cortices show reductions in PSD-95 protein levels (*Figure 2*). This finding is consistent with the decrease in mature dendritic spine characteristics in $Ppt1^{-/-}$ neurons in vitro and in vivo (*Figures 4* and *6*). However, the lack of PPT1 function had no direct effect on PSD-95 palmitoylation state (*Figure 11A*). Thus, we argue that the perturbed GluN2B to GluN2A switch is primarily responsible for excessive filopodial formation in $Ppt1^{-/-}$ neurons. While underrepresented PSD-95 expression correlates with impaired spine maturation, this is likely a secondary effect. Further study is warranted to identify the PPT1 substrates that directly regulate spinogenesis.

## Implications for other neurodegenerative diseases

While substantial progress has been made in our understanding of adult-onset neurodegenerative diseases including Alzheimer's disease and Parkinson's disease, effective, disease-modifying therapeutics are yet to be developed for most of these disorders. In part, this is likely due to the genetic complexity and heterogeneity of these diseases as well as lifestyle and environmental factors limiting the translational success of seemingly promising therapeutic strategies. Recently, studies in

monogenic diseases have attracted attention because they share common pathological hallmarks with adult-onset neurodegenerative diseases, including lipofuscin. This approach has turned out to be valuable to decipher underlying disease mechanisms in Parkinson's disease, for instance (*Peltonen et al., 2006*; *Neudorfer et al., 1996*; *Tayebi et al., 2001*; *Sidransky et al., 2009*; *Sidransky and Lopez, 2012*).

Our data indicate a significant dysregulation of NMDAR composition and function in the *Ppt1*$^{-/-}$ cortex associated with GluN2B and Fyn hyperpalmitoylation. Importantly, GluN2B has already been implicated in psychiatric and neurodegenerative disorders, including Alzheimer's disease (*Paoletti et al., 2013*; *Yamamoto et al., 2015*). Furthermore, Fyn is currently being investigated in clinical trials for Alzheimer's disease (*Nygaard et al., 2014*; *Nygaard et al., 2015*; *Kaufman et al., 2015*). Hence, our findings in CLN1 corroborate evidence in adult-onset neurodegenerative disorders and converge on disruption of Fyn kinase, GluN2B, or both as a shared feature of neurodegeneration. GluN2B and Fyn function therefore represent promising therapeutic targets for CLN1 and beyond.

The importance of palmitoylation at the synapse imply that additional mechanisms linking dysregulated protein palmitoylation to neurological diseases will likely be revealed. For instance, AMPAR and GABAR subunits undergo palmitoylation (*Hayashi et al., 2005*; *Fang et al., 2006*) and recent work demonstrates that deficient AMPAR palmitoylation facilitates seizure activity in vivo (*Itoh et al., 2018*). Further, preliminary results from our lab show developmental hyperpalmitoylation of at least one AMPAR subunit in *Ppt1*$^{-/-}$ animals (Koster, unpublished findings). Hence, the regulation of these receptors may also be involved in the pathogenesis of CLN1 or other diseases with perturbations in the balance between palmitoylation and depalmitoylation. Importantly, the palmitoylation of amyloid precursor protein (APP) and huntingtin are implicated in Alzheimer's disease and Huntington's disease pathogenesis, respectively (*Huang et al., 2004*; *Smith et al., 2005*; *Zheng and Koo, 2006*; *Bhattacharyya et al., 2013*). Thus, our results extend a growing body of evidence implicating protein palmitoylation in neurological diseases and warrant further investigation of protein depalmitoylation as a therapeutic target.

# Materials and methods

**Key resources table**

| Reagent type (species) or resource | Designation | Source or reference | Identifiers | Additional information |
|---|---|---|---|---|
| Strain, strain background (*Mus musculus*) | B6;129-Ppt1$^{tm1Hof}$/J | Jax stock #: 004313 | *Gupta et al., 2001*; RRID:MGI:004313 | |
| Antibody | Rabbit polyclonal anti-GluN2A | Novus Biologicals | Cat: NB300-105; RRID:AB_10001400 | (1:1000) |
| Antibody | Mouse monoclonal anti-GluN2B | UC Davis/NIH NeuroMab Facility | Cat: 75/097; RRID:AB_10673405 | (1:1000) |
| Antibody | Mouse monoclonal anti-GluN1 | UC Davis/NIH NeuroMab Facility | Cat: 75/272; RRID:AB_11000180 | (1:1000) |
| Antibody | Mouse monoclonal anti-PSD-95 | UC Davis/NIH NeuroMab Facility | Cat: K28/74; RRID:AB_2315909 | (1:2000) |
| Antibody | Mouse monoclonal anti-SAP102 | UC Davis/NIH NeuroMab Facility | Cat: N19/2; RRID:AB_2261666 | (1:2000) |
| Antibody | Rabbit polyclonal anti-Fyn | Cell Signaling | Cat: 4032 | (1:1000) |
| Antibody | Rabbit polyclonal anti-PPT1 | Gift from Sandra Hofmann | | (1:500); Dr. Hofmann |
| Antibody | Mouse monoclonal anti-β-actin-HRP | ThermoFisher Scientific | Cat: MA5-15739-HRP; RRID:AB_2537667 | (1:2000) |
| Antibody | Rabbit polyclonal anti-MAP2 | Millipore Sigma | Cat: AB5622; RRID:AB_91939 | (1:400) |

*Continued on next page*

*Continued*

| Reagent type (species) or resource | Designation | Source or reference | Identifiers | Additional information |
|---|---|---|---|---|
| Antibody | Rat monoclonal anti-LAMP2 | abcam | Cat: ab13524; RRID:AB_2134736 | (1:400) |
| Recombinant DNA reagent | pEF-GFP | Addgene | Plasmid: 11154 | Drs. Matsuda and Cepko |
| Recombinant DNA reagent | G-CaMP3 | Addgene | Plasmid: 22692 | Dr. Looger |
| Commercial assay or kit | PrestoBlue cell viability assay | ThermoFisher Scientific | Cat: A13261 | |
| Chemical compound, drug | 2-bromopalmitate | Sigma | Cat: 238422 | Treatment: 1 μM |
| Chemical compound, drug | cerulenin | Cayman Chemicals | Cat: 10005647 | Treatment: 1 μM |
| Software, algorithm | Fiji | | | |

## Animals, group allocation, and data handling

All animal procedures were performed in accordance with the guidelines of the University of Illinois of Chicago Institutional Animal Care and Use Committee. $Ptt1^{+/-}$ (heterozygous) mice were obtained from Jackson Laboratory and maintained on 12 hr light/dark cycle with food and water *ad libitum.* Breeding of $Ptt1^{+/-}$ ± results in litters containing $Ptt1^{-/-}$, $Ptt1^{+/-}$, and $Ptt1^{+/+}$ (WT) animals. $Ptt1^{-/-}$ and WT littermate controls at specified developmental time points: P11, P14, P28, P33, P42, P60, P78, and P120 were genotyped in-house (Gupta et al., 2001) and used for experiments. Although we used the littermate control system, in which WT and $Ptt1^{-/-}$ mice from the same litters were compared, each n was treated independently in statistical testing (pair-wise tests were not used). Imaging data was acquired randomly for each experiment (no criteria for selecting cells, view fields, etc. except where anatomically necessary, e.g. Figure 1). All data were acquired and maintained without descriptive naming/labeling to ease randomization. Data was randomized by students within the lab prior to analysis by KPK.

## Brain fractionation and immunoblotting

For collection of brain for biochemistry (immunoblot), $Ptt1^{-/-}$ and WT animals were decapitated following isoflurane anesthesia, then the brain was removed, and washed in ice cold PBS. The occipital cortex (visual cortex), hippocampus, and remaining cortex were separately collected on ice. Isolated visual cortices from $Ptt1^{-/-}$ and WT animals were homogenized in ice-cold synaptosome buffer (320 mM sucrose, 1 mM EDTA, 4 mM HEPES, pH7.4 containing 1x protease inhibitor cocktail (Roche), 1x phosphatase inhibitor cocktail (Roche) and 1 mM PMSF) using 30 strokes in a Dounce homogenizer. Aliquots for whole lysate (WL) were stored and the remaining sample was used for synaptosome preparation, performed as previously with slight modification. In brief, WLs were centrifuged at 1000 x g to remove cellular debris, supernatant was then centrifuged at 12,000 x g for 15 min to generate pellet P2. P2 was resuspended in synaptosome buffer and spun at 18,000 x g for 15 min to produce synaptosomal membrane fraction, LP1, which was used for downstream biochemical analyses (synaptosomes). For immunoblot, protein concentration of each sample was determined using BCA protein assay (Pierce). Samples were then measured to 20 μg total protein in 2x Laemmli buffer containing 10% β-mercaptoethanol (Bio-rad), boiled at 70°C for 10 min and loaded into 10% tris-glycine hand cast gels (Bio-rad), or 4–20% precast gels (Bio-rad) for electrophoresis (110V, 1.5–2 hr). Proteins were wet-transferred to PVDF membranes (Immobilon-P, Millipore), blocked in TBS, pH7.4 containing 5% non-fat milk and 0.1% Tween-20 (TBS-T +5% milk). Membranes were incubated in primary antibody solutions containing 2% BSA in TBS-T for 2 hr at RT or overnight at 4°C. Primary antibodies were used as follows: GluN2A (Cat: NB300-105, 1:1,000, Novus Biologicals), GluN2B (Cat: 75/097, 1:1,000, Neuromab), GluN1 (Cat: 75/272, 1:1000, Neuromab), PSD-95 (Cat: K28/74, 1:2,000, Neuromab), SAP102 (Cat: N19/2, 1:2,000, Neuromab), Fyn kinase (Cat: 4023, 1:1,000, Cell signaling), PPT1 (kindly provided by Dr. Sandra Hofmann, and β-actin-HRP (Cat: MA5-15739-HRP, 1:2,000, ThermoFisher). Membranes were then incubated with appropriate secondary, HRP-conjugated antibodies

(Jackson ImmunoResearch) at either 1:5,000, 1:10,000, or 1:30,000 (PSD-95 only) for 1 hr at RT. Visualization and quantification was performed using Pierce SuperSignal ECL substrate and Odyssey-FC chemiluminescent imaging station (LI-COR). Signal density for each synaptic protein was measured using the LI-COR software, Image Studio Lite (version 5.2) and was normalized to the signal density for β-actin loading control for each lane. A total of four independent experiments was performed for both WL and LP1 analyses, with a minimum of two technical replicates for each experiment averaged together.

## Histology and autofluorescent lipopigment quantification

$Ppt1^{-/-}$ and WT mice were anesthetized using isoflurane and transcardially perfused with ice cold PBS (pH 7.4,~30 ml/mouse) followed by 4% paraformaldehyde (PFA) in PBS (~15 ml/mouse). Brains were removed and post-fixed for 48 hr at 4°C in 4% PFA and transferred to PBS, pH7.4 containing 0.01% sodium azide for storage if necessary. Brains from $Ppt1^{-/-}$ and WT animals were incubated in 30% sucrose solution for 48 hr prior to sectioning using Vibratome 1000 in cold PBS. For imaging and quantification of AL, sagittal sections were cut at 100 μm. Every third section was mounted on Superfrost Plus microscope slides (VWR) using Vectamount mounting media containing DAPI (Vector Laboratories, cat: H-5000). Interlaced/overlapping images of visual cortex area V1 from the cortical surface to subcortical white matter (or subiculum), which was localized using Paxino's mouse atlas (sagittal), were collected for 2–4 sections from each animal using a Zeiss LSM710 confocal laser scanning microscope at 40x magnification (excitation at 405 nm to visualize DAPI and 561 nm to visualize AL). All sections were imaged using identical capture conditions. Quantification of AL was performed by thresholding images in FIJI (NIH), generating a binary mask of AL-positive pixels (satisfied threshold) vs. background. The identical threshold was applied to each image (from cortical surface to subcortical white matter and across animals). Percent area occupied by AL puncta that satisfied the threshold was then calculated using the 'analyze particles' tool in FIJI. This analysis was performed for 2–4 sections (total of ~10–20 images, as imaging an entire cortical column is typically five interlaced images) from each animal and averaged together to give a single value, representative of the total area occupied by AL in the cortical column imaged. Three to six animals per group were analyzed this way and averaged to give the mean area occupied by AL at each time point, for both genotypes (n = 4–6 animals/group).

## Electrophysiology

WT and $Ppt1^{-/-}$ animals at P42 were deeply anesthetized using isoflurane drop method and decapitated. Brains were resected in semi-frozen oxygenated (95% $O_2$ and 5% $CO_2$) artificial cerebrospinal fluid (aCSF, in mM: NaCl 85, sucrose 75, KCl 2.5, $CaCl_2$ 0.5, $MgCl_2$ 4, $NaHCO_3$ 24, $NaH_2PO_4$ 1.25, D-glucose 25, pH 7.3), and 350 μm sections containing visual cortex area V1 were sectioned using a Leica VT1200 S vibratome in semi-frozen aCSF. After recovery (1 hr) in aCSF at 30°C, sections were transferred to the recording chamber, perfused at 2 ml/min with aCSF at 30°C. Following localization of visual cortex area V1 using Paxinos mouse brain atlas, a stimulating electrode was placed in layer IV, and pyramidal neurons from layer II/III were blindly patched (patch solution in mM: CsOH monohydrate 130, D-Gluconic acid 130, EGTA 0.2, $MgCl_2$ 1, CsCl 6, Hepes 10, $Na_2$-ATP 2.5, Na-GTP 0.5, Phoshocreatine 5, QX-314 3; pH 7.3, osmolarity 305 mOsm) and recorded in voltage clamp mode at +50 mV ($V_H$) to remove $Mg^{2+}$ block from NMDARs. NMDA-EPSCs were pharmacologically isolated via addition of CNQX (10 μM), (+)-Bicuculline (60 μM) and SCH 50911 to block AMPA, $GABA_A$ and $GABA_B$ receptors, respectively. Stimulation intensity was titrated to give a saturating postsynaptic response, and EPSCs were then recorded, averaging 5–10 sweeps. The decay phase of the averaged NMDAR-EPSCs were then fitted to a double exponential (*Stocca and Vicini, 1998*; *Vicini et al., 1998*). We calculated for each cell: the amplitude of the fast (Af) component (GluN2A-mediated), the amplitude of the slow (As) component (GluN2B-mediated), the contribution of the fast component Af/Af + As to the overall decay phase, the τ fast (τf), the τ slow (τs) and the τ weighted (τw) in WT and $Ppt1^{-/-}$ mice following this formula: τw= τfx(Af/Af +As) + τsx(As/Af +As) (n = 8/4 (cells/animals), WT; n = 8/5 PPT-KO). For experiments in *Figure 3E*, baseline NMDAR-EPSCs were recorded as typical and compared to those following 30 min of bath-infused Ro 25–6981 (3 μM).

## In utero electroporation

In utero electroporation was performed as previously described (*Yoshii et al., 2011*). Timed-pregnant dams at E16.5 were deeply anesthetized via isoflurane (3% induction, 1–1.5% for maintenance of anesthesia during surgery) and laparotomized. The uterus was then externalized and up to ~1 μl of solution containing GFP construct (2 μg/μl) and fast green dye was delivered into the left lateral ventricle through the uterine wall using a micropipette. Using an ECM 830 Square Wave electroporator (Harvard Apparatus, Holliston MA), brains were electroporated with 5 pulses of 28V for 50 ms at intervals of 950 ms at such an angle to transfect neurons in visual cortex. After recovery, pregnancies were monitored, and pups were delivered and nursed normally. Electroporated pups were genotyped, raised to P33, and sacrificed via transcardial perfusion as described above. Electroporated brains from WT and *Ppt1*-/- mice (procedure schematized in *Figure 4A*) were sectioned and sequentially mounted. Electroporated neurons in visual cortex (*Figure 4B*) were imaged to capture all apical neurites and 3D reconstructed images were analyzed in Imaris (Bitplane) for dendritic spine characteristics known to be associated with synaptic maturity (spine density, spine length, spine volume, and spine head volume). At least two z-stack images (typically >100 z-planes/image) were stitched together to capture the prominent apical neurites and extensions into the cortical surface for each cell. Each stitched image, equivalent to one cell, was considered one n.

## Primary cortical neuron culture

For primary cortical neuron cultures, embryos from timed-pregnant, *Ppt1*-/+ dams were removed, decapitated, and cortices resected at embryonic day (E) 15.5. All dissection steps were performed in ice cold HBSS, pH7.4. Following cortical resection, tissue from each individually-genotyped embryo were digested in HBSS containing 20 U/ml papain and DNAse (20 min total, tubes flicked at 10 min) before sequential trituration with 1 ml (~15 strokes) and 200 μl (~10 strokes) pipettes, generating a single-cell suspension. For live-cell imaging experiments, cells were counted then plated at 150,000–180,000 cells/well in 24-well plates containing poly-D-lysine/laminin-coated coverslips. For biochemical experiments, that is immunoblot, APEGS assay in vitro, cells were plated on poly-D-lysine/laminin-coated 6-well plates at 1,000,000 cells/well. Cells were plated and stored in plating medium (Neurobasal medium containing B27 supplement, L-glutamine and glutamate) for 3–5 DIV, before replacing half medium every 3 days with feeding medium (plating medium without glutamate). Cultures used in chronic palmitoylation inhibitor treatment were exposed to either DMSO (vehicle), 2 BP (1 μm, Sigma, cat: 238422) or cerulenin (1 μm, Cayman Chemicals, cat: 10005647) every 48 hr between DIV 12 and 18.

## Primary cortical neuron harvest and immunoblotting

Primary cortical neurons from E15.5 WT and *Ppt1*-/- embryos were cultured for 7, 10, or 18 DIV prior to harvest for immunoblot or APEGS assay (only DIV18 used for APEGS). To harvest protein extracts, cells were washed 2x with ice-cold PBS before addition of lysis buffer containing 1% SDS and protease inhibitor cocktail, 500 μl/well. Cells were incubated and swirled with lysis buffer for 5 min, scraped from the plate, triturated briefly, and collected in 1.5 ml tubes. Lysates were centrifuged at 20,000 g for 15 min to remove debris, and the supernatant was collected for biochemical analysis. Immunoblotting analyses were performed as above. APEGS assay was carried out as described in the following section.

## APEGS assay on primary cortical neuron lysates

The APEGS assay was performed as utilized in (*Yokoi et al., 2016*) and recommended by Dr. M. Fukata (personal communication, 06/2018). Briefly, cortical neuron lysates were brought to 150 μg total protein in a final volume of 0.5 ml buffer A (PBS containing 4% SDS, 5 mM EDTA, protease inhibitors, remaining sample used in aliquots for 'input'). Proteins were reduced by addition of 25 mM Bond-Breaker TCEP (0.5M stock solution, ThermoFisher) and incubation at 55°C for 1 hr. Next, to block free thiols, freshly prepared N-ethylmaleimide (NEM) was added to lysates (to 50 mM) and the mixture was rotated end-over-end for 3 hr at RT. Following 2x chloroform-methanol precipitation (at which point, protein precipitates were often stored overnight at −20°C), lysates were divided into +hydroxylamine (HA) and −HA groups for each sample, which were exposed to 3 volumes of HA-containing buffer (1 M HA, to expose palmitoylated cysteine residues) or Tris-buffer control (-

HA, see *Figure 10*), respectively, for 1 hr at 37°C. Following chloroform-methanol precipitation, lysates were solubilized and exposed to 10 mM TCEP and 20 mM mPEG-5k (Laysan Bio Inc., cat# MPEG-MAL-5000–1 g) for 1 hr at RT with shaking (thereby replacing palmitic acid with mPEG-5K on exposed cysteine residues). Following the final chloroform-methanol precipitation, samples were solubilized in a small volume (60 μl) of PBS containing 1% SDS and protein concentration was measured by BCA assay (Pierce). Samples were then brought to 10 μg protein in laemmli buffer with 2% β-mercaptoethanol for immunoblot analyses as above. Quantification of palmitoylated vs. non-palmitoylated protein was carried out as above, with the added consideration that palmitoylated protein was taken as the sum of all (typically two-three distinct bands, see *Figure 9*) bands demonstrating the APEGS-dependent molecular weight shift compared to the –HA control lane. Non-palmitoylated protein was quantified from the band size-matched to the –HA control sample. The ratio was taken as the palmitoylated protein divided by non-palmitoylated protein, all divided by β-actin control from the same lane.

## Transfection, dendritic spine and calcium imaging analyses

For analysis of dendritic spine morphology, WT and *Ppt1*[-/-] neurons were transfected between DIV6-8 with GFP using Lipofectamine 2000 (ThermoFisher) according to manufacturer protocol with a slight modification. Briefly, GFP DNA construct (~2 μg/μl, added at ~1 μg/well) was mixed with Lipofectamine-containing Neurobasal medium, incubated for 30 min to complex DNA-Lipofectamine, equilibrated to 37°C, and added to the cells 250 μl/well for 1–1.5 hr. Following incubation, complete medium was returned to the cells. Neurons were then imaged at DIV15 and DIV20 for dendritic spine morphology using a Zeiss LSM 710 confocal microscope equipped with a heated stage at 63x magnification. GFP-positive neurons were imaged at 0.2 μm Z-plane interval (typically 25 Z-planes/image). Three to seven overlapping Z-stacks were stitched to visualize an entire neuron. Z-stack images were collapsed into a single plane and dendritic spines were analyzed using semi-automated image processing software, Imaris (Bitplane). The same dendrite and dendritic spine processing parameters were used for each image. For DIV15: n = 4–5 neurons/group, three-independent experiments, WT = 21,514 spines; *Ppt1*[-/-] = 18,013 spines. For DIV20: n = 3 neurons/group, two-independent experiments, WT = 11,335 spines; *Ppt1*[-/-] = 9958 spines.

To directly image calcium signals in WT and *Ppt1*[-/-] neurons, cells were transfected as above using the construct encoding GCaMP3 (see Acknowledgments) at DIV8. A subset of cells (*Figure 10*) were treated with 2 BP (1 μM) or cerulenin (1 μM) from DIV12-18. Cells were grown to DIV18 then imaged at room temperature in Tyrode's solution (imaging medium, 139 mM NaCl, 3 mM KCl, 17 mM NaHCO$_3$, 12 mM glucose, and 3 mM CaCl$_2$) for a maximum of 15 min using a Mako G-507B camera mounted onto a Leica inverted microscope. Videos were acquired at ~7 framess using StreamPix software (NorPix). A maximum of 5 min per neuron was recorded (thus, minimum three neurons per coverslip were acquired). N = 3–6 neurons/group, three independent experiments. For treatment with Ro 25–6981, neurons were imaged at baseline for 2–2.5 min before adding Ro 25–6981 (1 μM) directly to the imaging medium. Neurons were then imaged for an additional 2.5 min.

To analyze the area under the curve (AUC) and width (diffusion distance) of calcium transients, 500–600 frames from the middle of each video (average frame count for whole videos=~2200 frames) for WT, *Ppt1*[-/-], and *Ppt1*[-/-] palmitoylation inhibitor-treated (treatment performed as in 'Primary cortical neuron culture section; DIV12-18, every other day, 1 μM) neurons were analyzed using FIJI (NIH). Dendritic segments, excluding primary dendrites, were traced using a segmented line ROI with pixel width of 50, which reliably encompassed the dendritic segment and accompanying dendritic spines. Next, the following macro derived from the ImageJ forum (http://forum.imagej.net/t/how-to-obtain-xy-values-from-repeated-profile-plot/1398) was run on each individual ROI:

```
macro 'Stack profile Plot' {
    collectedValues="";
    ymin = 0; ymax = 255;
    saveSettings();
    if (nSlices == 1)
        exit('Stack required');
    run('Profile Plot Options...',
        'width = 400 height = 200 minimum='+ymin + ' maximum='+ymax + ' fixed');
```

```
setBatchMode(true);
stack1 = getImageID;
stack2 = 0;
n = nSlices;
for (i = 1; i <= n; i++) {
    showProgress(i, n);
    selectImage(stack1);
    setSlice(i);
    run('Clear Results');
    profile = getProfile();
    for (j = 0; j < profile.length; j++) {
        collectedValues = collectedValues + profile[j] + '\t';
    }
    collectedValues = collectedValues + '\n';
    run('Plot Profile'); run('Copy');
    w = getWidth;
    h = getHeight;
    close();
    if (stack2 == 0) {
        newImage('Plots', '8-bit', w, h, 1);
        stack2 = getImageID;
    } else {
        selectImage(stack2);
        run('Add Slice');
    }
    run('Paste');
}
f = File.open('C:/'cell#, ROI #".xls');
print(f, collectedValues);
setSlice(1);
setBatchMode(false);
restoreSettings();
}
```

This gives the fluorescence intensity at each pixel along the ROI across the time/frame dimension. The background fluorescence for each ROI was then subtracted by averaging the fluorescence across the ROI in an inactive state (no calcium transients), giving the measure $\Delta F/F_0$ when examined across time/frame. For each ROI (up to 1265 pixels in length), each calcium transient at individual synaptic sites (dendritic spines or dendritic shafts) was averaged. Those averages were then compiled to give the average transient signal, which was then used to analyze the AUC and calcium diffusion distance (n = 3 neurons/group/experiment, three distinct cultures: WT = 55 ROIs, 185 synaptic sites, 1630 transients; $Ppt1^{-/-}$ = 38 ROIs, 131 synaptic sites, 1281 transients; $Ppt1^{-/-}$ + 2-BP = 28 ROIs, 82 synaptic sites, 420 transients; $Ppt1^{-/-}$ + cerulenin = 24 ROIs, 82 synaptic sites, 540 transients). For Ro 25–6981-treated neurons, the same protocol was followed with the exception that calcium transients at an individual synaptic site were split into 'before application' and 'after application' groups.

To analyze synaptic synchrony, $\Delta F/F_0$ measurements for 20 randomly-chosen sites of synaptic activity per neuron were correlated across the time dimension (500 frames of each video). A correlation matrix was generated to determine the average correlation of each synaptic site with all other chosen sites. The average values for each synaptic site, for five neurons/group are plotted in *Figure 7*.

## NMDA toxicity assays

To measure cell viability following exposure of WT and $Ppt1^{-/-}$ neurons to NMDA and glycine, neurons were plated as above and grown to DIV18. For experiments presented in *Figure 6*, feeding medium was removed from neurons, stored at 37°C, and replaced with B27-free Neurobasal medium

with or without NMDA/glycine at the following concentrations: 10/1 µM, 100/10 µM, or 300/30 µM (ratio maintained at 10:1). Cells were incubated for 2 hr at 37°C in treatment medium. Following incubation, treatment medium was removed and replaced with the original feeding medium. Cells were then incubated an additional 22 hr before addition of PrestoBlue cell viability reagent (Thermo-Fisher). At 24 hr, fluorescence intensity of each well was measured using a Beckman Coulter DTX 800 Multimode Detector. Cell viability for each treatment condition was calculated and expressed as percentage of vehicle-treated control wells (no pretreatment, no NMDA application). Experiments in *Figure 7* were performed similarly except that cultures were pretreated with either DMSO (vehicle), 2 BP (1 µM, Sigma, cat: 238422) or cerulenin (1 µM, Cayman Chemicals, cat: 10005647) every 48 hr between DIV 11 and 18.

## AL accumulation in vitro, palmitoylation inhibitor treatment, imaging and analysis

WT and $Ppt1^{-/-}$ neurons were cultured as above. To examine AL deposition, neurons were grown to DIV18-20, fixed in 4% PFA for 10 min at RT, and stored in PBS for up to 72 hr prior to immunocytochemistry. To examine AL accumulation alone, cells were immunostained for the microtubule associated protein, MAP2 (Millipore Sigma, cat: AB5622) and mounted in DAPI-containing mounting medium. To assess AL localization, DIV18-DIV20 neurons were immunostained for MAP2 and LAMP-2 (Abcam, cat: ab13524). Neurons were then imaged at random using a Zeiss LSM 710 confocal microscope at 63x magnification. Z-stacks (0.4 µm Z-plane interval, 12–22 Z-planes/image) were taken at 512 × 512 pixel density. 7–10 neurons/group for three independent experiments.

To semi-automatically analyze the percentage of AL-containing cells, the cytosolic area covered by AL deposits, and the cytosolic area covered by lysosomes, images immunostained for MAP2 and LAMP-2 were processed in FIJI. Each channel of the image: LAMP-2 (488 nm), MAP2 (633 nm), DAPI (405 nm), AL (561 nm) was thresholded separately as to display only the lysosomes, cell soma, the nucleus, and AL deposits, respectively. Thresholds were kept identical between images. Next, the areas of these compartments/deposits were measured using the 'analyze particles' tool restricted to an ROI tracing the cell soma. Lysosomes needed to have a circularity of >0.5 to avoid counting small clusters of lysosomes as a single unit (*Bandyopadhyay et al., 2014*; *Grossi et al., 2016*). To measure AL deposits, the same approach was used with the additional constraint: AL deposits were required to have a circularity >0.4 and comprise more than eight adjacent pixels. Cytosolic area was calculated by measuring MAP2 signal area and subtracting the area occupied by DAPI stain.

For line scan analysis of the representative, vehicle treated WT and $Ppt1^{-/-}$ images in *Figure 9—figure supplement 1*, images were loaded in Fiji and channels split individually. Next, line scans of the cell soma across areas encompassing somatic lysosomes and, in the case of $Ppt1-/-$ neurons, AL deposits, were drawn manually as a line ROI. The 'plot profile' tool was then used to obtain the grey scale values for the fluorescence intensity of each channel across the same ROI. These values were then plotted either individually (as in B and E) or plotted with LAMP2 and AL signals overlapping (as in C and F).

## Immunocytochemistry

Coverslips were stained in runs so that all experimental and control groups were immunostained simultaneously. Coverslips were washed 3x with TBS, permeabilized for 20 min at RT with TBS containing 0.5% Triton X-100 and blocked for 1 hr at RT in TBS containing 0.1% Triton X-100% and 5% BSA. Then, primary antibody (MAP2 or LAMP-2) at 1:400 dilution was added to coverslips in TBS containing 0.1% Triton X-100% and 1% BSA and incubated for 2 hr at RT or overnight at 4°C. Following 4X washes with TBS containing 0.1% Triton X-100, cells were incubated with 1:400 secondary, fluorophore-linked antibody (either Alexa Fluor 488, cat. #: A-11034, A-11006; or Alexa Fluor 633, ThermoFisher, cat. #: A-21070) in TBS containing 0.1% Triton x-100% and 1% BSA. These steps are repeated for double immunostained cells. For LAMP-2/MAP2 double immunostaining, saponin was used in place of Triton X-100 at the same concentrations. Coverslips are then mounted on Super-Frost Plus slides in DAPI Vectamount medium.

## Acknowledgements

The authors thank Dr. Sandra Hofmann (Univ. of Texas Southwestern) for an anti-PPT1 antibody, and Dr. Froylan Calderon de Anda (Universitätsklinikum Hamburg-Eppendorf) for the GCaMP3 construct used to visualize calcium activity in *Figure 6*. This work is supported by startup funding awarded to AY by the University of Illinois at Chicago, Department of Anatomy and Cell Biology.

## Additional information

### Funding

| Funder | Author |
| --- | --- |
| University of Illinois at Chicago | Akira Yoshii |

The funders had no role in study design, data collection and interpretation, or the decision to submit the work for publication.

### Author contributions

Kevin P Koster, Conceptualization, Data curation, Formal analysis, Investigation, Methodology, Writing—original draft, Writing—review and editing; Walter Francesconi, Fulvia Berton, Formal analysis, Investigation, Methodology, Writing—review and editing; Sami Alahmadi, Data curation, Formal analysis, Validation; Roshan Srinivas, Data curation, Software, Formal analysis; Akira Yoshii, Conceptualization, Resources, Supervision, Funding acquisition, Investigation, Writing—original draft, Writing—review and editing

### Author ORCIDs

Kevin P Koster (iD) http://orcid.org/0000-0003-2935-3427
Akira Yoshii (iD) http://orcid.org/0000-0001-8305-006X

### Ethics

Animal experimentation: All animal procedures were performed in accordance with the guidelines of the University of Illinois of Chicago Institutional Animal Care and Use Committee. All animals were handled and treated as outlined under the Institutional Animal Care and Use Committee (IACUC) protocol (#17-209). All efforts were made to minimize animal suffering.

### Decision letter and Author response

Decision letter https://doi.org/10.7554/eLife.40316.025
Author response https://doi.org/10.7554/eLife.40316.026

## Additional files

### Supplementary files

• Supplementary file 1. Mean values, s.e.m., and n for the bar chart in *Figure 1C*. Values are represented for each layer, at each age in WT and $Ppt1^{-/-}$ mice.
DOI: https://doi.org/10.7554/eLife.40316.022

• Transparent reporting form
DOI: https://doi.org/10.7554/eLife.40316.023

### Data availability

All data generated or analysed during this study are included in the manuscript.

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
