## [Decision Letter]

Thank you for submitting your article "NMDA Receptor Dysregulation by Defective Depalmitoylation in the Infantile Neuronal Ceroid Lipofuscinosis Mouse Model" for consideration by *eLife*. Your article has been reviewed by three peer reviewers, one of whom is a member of our Board of Reviewing Editors, and the evaluation has been overseen by a Reviewing Editor and a Senior Editor. The reviewers have opted to remain anonymous.

The reviewers have discussed the reviews with one another and the Reviewing Editor has drafted this decision to help you prepare a revised submission.

This study addresses a role for depalmitoylation enzyme, PPT1, whose mutations are associated with ceroid lipofuscinosis (CLN1). Specifically, the authors examine PPT1 functions in properties associated with synapse maturation in the mouse visual cortex and cortical cultures. They find that the loss of PPT1 is accompanied by altered spine morphology, compromised developmental switch of GluN2B to GluN2A-NMDARs, and enhanced Ca^2+^ signaling and vulnerability to excitotoxicity. Notably, the hyperpalmitoylation of GluN2B and excitotoxicity in PPT1 KO neurons can be attenuated by chronic block of protein palmitoylation, suggesting the central role for aberrant palmitoylation of GluN2B in CLN1. Overall, the main findings are novel and exciting, and the study should be of general interest. However, all three reviewers have a major criticism concerning the fact that the link between the observed lack of change in the total level of GluN2B, the palmitoylation state of GluN2B and how it affects the synaptic NMDAR subunit composition and excitotoxicity remains weak. These points need to be better established with additional experiments that are listed below, which should be feasible to be completed within two months.

Essential revisions:

1) Similarly to GluN2B experiments shown, the authors need to test for palmitoylation state of GluN2A and PSD-95 in WT, PPT1 KO and PPT1 KO treated with palmitoylation inhibitors in cultured neurons. Notably, previous work from the senior author has shown that activity-dependent palmitoylation of PSD-95 promotes synapse maturation and implicated the presence of different depalmitoylation enzymes that act on PSD-95. Therefore, it will be important to test the level of palmitoylation of PSD-95 and GluN2A in PPT1 KO mice to clarify whether different palmitoylation-depalmitoylation pathways are specific to selective synaptic targets. Additionally, three independent experiments will be needed for the treatment. The results should be discussed in light of decreased PSD-95 levels in synaptoneurosome in PPT1 KO mice.

2) That PPT1 specifically depalmitoylates GluN2B but not GluN2A should be tested directly. Otherwise the authors need to tone down the claim of direct depalmitoylation by PPT1 and discuss the possibility that PPT1 might be indirectly affecting the palmitoylation state of GluN2B.

3) It will be important to show the developmental profile of PPT1 in WT visual cortex, and check whether the expression profile correlates to the timing of accumulation of AL deposits and synapse phenotype seen in the KO animals. The result should be used to rationalize and discuss about the late onset of the biochemical changes at P33 onwards. Although the cited papers (Suopanki et al., 1999a, 1999b) attempted to describe the profile, the sample sizes were too low (n = 1) and the results were inconsistent to draw any meaningful conclusion. Moreover, the authors state that, "We found that ALs are detectable first at P14 in *Ppt1^-/-^* visual cortex, much earlier than previously reported (Figure 1A, B, C)." The authors should state the age at which ALs were previously demonstrated and cite the appropriate reference.

4a) The analysis of EPSC waveforms as shown is not compelling. The altered subunit composition of synaptic NMDAR currents in PPT1 KO neurons and its reversal by treatment with palmitoylation inhibitors, should be demonstrated using GluN2A or GluN2B subunit-specific blockers.

4b) Are there changes in synaptic AMPAR currents in PPT1 KO neurons?

4c) In order to support the reversal of excitotoxicity in PPT1 KO neurons following treatment with palmitoylation inhibitors, the reversal of altered Ca^2+^ transients should also be demonstrated under the same treatment conditions.

5) Figures 4 and 6 should include comparisons of spine density.

6) Figure 9F. Control for 2-BP treatment (in the absence of NMDA+glycine) is missing.

[Editors' note: further revisions were requested prior to acceptance, as described below.]

Thank you for submitting your revised article "NMDA Receptor Dysregulation by Defective Depalmitoylation in the Infantile Neuronal Ceroid Lipofuscinosis Mouse Model" for consideration by *eLife*. Your article has been reviewed by three peer reviewers, one of whom is a member of our Board of Reviewing Editors, and the evaluation has been overseen by a Reviewing Editor and Huda Zoghbi as the Senior Editor. The reviewers have opted to remain anonymous.

The reviewers have discussed the reviews with one another and the Reviewing Editor has drafted this decision to help you prepare a revised submission.

Summary:

In the revised submission, the authors have addressed some of the essential concerns raised by the reviewers. Specifically, the authors have added (1) the developmental profile of PPT1, (2) control experiments for the culture work testing the effect of blocking palmitoylation on excitotoxicity in wild type and PPT1 KO neurons, and (3) spine density info in Figure 4 and 6. Despite some improvements, however, the revisions fall short of providing mechanistic insights into the link between PPT1 loss and NMDAR dysregulation in the CLN1 mouse model, and it is not clear how the palmitoylation state of GluN2B is related to the synaptic NMDAR subunit composition and excitotoxicity. Thus, the main conclusion that "depalmitoylation of GluN2B by PPT1 plays a critical role in postsynapse maturation and neurodegeneration" is not directly supported by the experimental evidence provided. At minimum, following essential revisions are required.

Essential revisions:

1) As requested in the original list of essential required revision, one needs to test if the palmitoylation state of PSD-95 and GluN2A is affected, regardless of whether PPT1 can directly depalmitoylate GluN2B.

2) Changes in synaptic AMPAR currents in PPT1 KO neurons should be reported. Such information is required in order to fully understand the extent to which the observed effects of PPT1 loss are specifically mediated by GluN2B.

3) The claim that the enhancement of slow synaptic NMDR current observed in PPT1 KO neurons is mediated specifically by NR2B-containing NMDARs, should be tested pharmacologically.

4) The reversal of excitotoxicity in PPT1 KO neurons by palmitoylation inhibitors should be supported by demonstrating also the reversal of altered Ca^2+^ transients under the same treatment conditions.

---

## [Author Response]

Essential revisions:1) Similarly to GluN2B experiments shown, the authors need to test for palmitoylation state of GluN2A and PSD-95 in WT, PPT1 KO and PPT1 KO treated with palmitoylation inhibitors in cultured neurons. Notably, previous work from the senior author has shown that activity-dependent palmitoylation of PSD-95 promotes synapse maturation and implicated the presence of different depalmitoylation enzymes that act on PSD-95. Therefore, it will be important to test the level of palmitoylation of PSD-95 and GluN2A in PPT1 KO mice to clarify whether different palmitoylation-depalmitoylation pathways are specific to selective synaptic targets. Additionally, three independent experiments will be needed for the treatment. The results should be discussed in light of decreased PSD-95 levels in synaptoneurosome in PPT1 KO mice.

Due to unforeseen circumstances, we were not able to obtain these data for the manuscript. However, we will continue to determine palmitoylation levels of PSD-95 and GluN2A in future and are willing to include the results in this manuscript if the other new data and analyses in this revision do not compellingly conclude that hyperpalmitoylation of GluN2B subunit perturbs GluN2B to GluN2A switch in *Ppt1^-/-^* mice.

2) That PPT1 specifically depalmitoylates GluN2B but not GluN2A should be tested directly. Otherwise the authors need to tone down the claim of direct depalmitoylation by PPT1 and discuss the possibility that PPT1 might be indirectly affecting the palmitoylation state of GluN2B.

Indeed, one future direction of our work is to determine *if* and how PPT1 depalmitoylates GluN2B directly or, alternatively, how lack of PPT1 indirectly regulates other mediators to achieve the observed effects. While we will surely aim to address these questions in future, we have substantially softened our mechanistic insights into the GluN2B depalmitoylation by PPT1. We have also added a section in the Discussion outlining potential mechanisms by which defective PPT1 may lead indirectly to GluN2B hyperpalmitoylation, with a focus on mechanisms involving Src family kinases such as Fyn (subsection “GluN2B to GluN2A subunit switch”, second paragraph).

3) It will be important to show the developmental profile of PPT1 in WT visual cortex, and check whether the expression profile correlates to the timing of accumulation of AL deposits and synapse phenotype seen in the KO animals. The result should be used to rationalize and discuss about the late onset of the biochemical changes at P33 onwards. Although the cited papers (Suopanki et al., 1999a, 1999b) attempted to describe the profile, the sample sizes were too low (n = 1) and the results were inconsistent to draw any meaningful conclusion. Moreover, the authors state that, "We found that ALs are detectable first at P14 in Ppt1^-/-^ visual cortex, much earlier than previously reported (Figure 1A, B, C)." The authors should state the age at which ALs were previously demonstrated and cite the appropriate reference.

We have included in Figure 2 and in Figure 2—figure supplement 1 the expression profile of PPT1 in the WT and *Ppt1^-/-^*visual cortex at the same ages examined for the other analyses. We demonstrate that in whole lysates (Figure 2—figure supplement 1D), PPT1 expression is suppressed until P28, peaks robustly at P33, and partially declines at P42 and onward. In synaptosomes (Figure 2E), however, PPT1 expression is low at P11 and P14, but steadily increases thereafter, with maximum expression at P60, the latest age examined. These data fit with the notion that PPT1 expression is linked to the regulation of GluN2 subunit composition during cortical development and match the timing of the observed reductions in GluN2A and PSD-95 in *Ppt1^-/-^*visual cortex. We have added the result in the text (subsection NMDAR subunit composition is biased toward immaturity in *Ppt1^-/-^* visual cortex”, last two paragraphs). The authors very much appreciate this suggestion and feel these data have substantially improved our manuscript.

We have also updated the manuscript with the appropriate citations for the mentioned statement.

4a) The analysis of EPSC waveforms as shown is not compelling. The altered subunit composition of synaptic NMDAR currents in PPT1 KO neurons and its reversal by treatment with palmitoylation inhibitors, should be demonstrated using GluN2A or GluN2B subunit-specific blockers.

After reviewing the data presented in Figure 3 and examining them in more detail, we discovered that scaled *Ppt1^-/-^*NMDAR currents (Figure 3C-E) demonstrate a detectably wider decay than those of WT cells, resulting in a significantly higher weighted decay constant (τw). This indicates more strongly that the contribution of the GluN2B component to the overall current is greater in *Ppt1^-/-^*cells compared to WT. Further, we detected a significantly increased NMDAR-EPSC rise time in *Ppt1^-/-^* responses, an indication of increased receptor distance from the presynaptic site and a characteristic of GluN2B-containing NMDARs (Townsend, 2003). The new analyses are now included in Figure 3 and in text (subsection “NMDAR-mediated EPSCs are altered in *Ppt1^-/-^* visual cortex”), Together, these data corroborate the notion that GluN2B is overrepresented in *Ppt1^-/-^*cells in vivo, and more directly fit with extrasynaptic Ca^2+^ entry in live-cell imaging data (Figure 7). We feel these novel analyses enhance the impact of the electrophysiological experiments significantly.

Unfortunately, we were unable to readily perform new electrophysiological experiments due to unforeseen circumstances following submission of the manuscript. However, we aim to resume performing ex vivo electrophysiological recordings in the future and will make this developmental profile a major focus of our experiments using GluN2-specific blockers. Lastly, we have a reservation about an acute treatment with palmitoylation inhibitors in ex vivoslice experiments. Our data indicate that the switching mechanism occurs over many days in vivo, thus chronic treatment would be required to suppress palmitoylation in vivo. However, blood brain barrier permeability of 2-BP and cerulenin are currently unknown. Without this information, peripheral treatment with these compounds in vivo would yield ambiguous results as poor brain penetration may lead to little or no effect on GluN2 palmitoylation. Thus, this experiment would need to be restructured (potentially requiring updates to our IACUC protocol, requiring intracerebroventricular injection, for example) or, more practically, deferred until next generation palmitoylation inhibitors become available.

4b) Are there changes in synaptic AMPAR currents in PPT1 KO neurons?

Indeed, we have recently obtained preliminary evidence to suggest that AMPAR currents are also altered in the *Ppt1^-/-^*visual cortex. However, we strongly feel that these AMPAR-related changes are beyond the scope of this manuscript, which specifically focuses on NMDAR regulation during development. Surely our future work will seek to understand the role of PPT1 in AMPAR regulation during development, homeostasis, and synaptic plasticity.

4c) In order to support the reversal of excitotoxicity in PPT1 KO neurons following treatment with palmitoylation inhibitors, the reversal of altered Ca^2+^ transients should also be demonstrated under the same treatment conditions.

Unfortunately, given the circumstances, we were unable to complete these experiments in the allotted time. We anticipate that, given the success of palmitoylation inhibitors in reducing GluN2B hyperpalmitoylation, that this treatment would also alleviate the aberrant calcium diffusion and perhaps reduce the frequency of postsynaptic responses in *Ppt1^-/-^* neurons (as the number of GluN2B-containing NMDARs would be reduced at the surface). However, this will have to be further examined in future.

5) Figures 4 and 6 should include comparisons of spine density.

Thank you for this suggestion. These data were encoded in the measures we gathered using the Imaris software. Interestingly, the data suggests that *Ppt1^-/-^*neurons have increased spine density compared to WT cells both in vivoand in vitro, indicating an aberrant increase in the number of synapses formed in *Ppt1^-/-^*cells. We have added the data to both Figures 4 and 6.

6) Figure 9F. Control for 2-BP treatment (in the absence of NMDA+glycine) is missing.

We have now displayed on the graph the% cell viability of neurons following treatment with 2-BP and cerulenin alone/without the addition of NMDA+glycine. These compounds at the doses utilized do not have a significant effect on neuronal survival and thus treatment with the drugs alone results in approximately 100% cell viability (see updated Figure 9F).

[Editors' note: further revisions were requested prior to acceptance, as described below.]

Essential revisions:1) As requested in the original list of essential required revision, one needs to test if the palmitoylation state of PSD-95 and GluN2A is affected, regardless of whether PPT1 can directly depalmitoylate GluN2B.

We have examined the palmitoylation state of PSD-95 and GluN2A in vehicle- or palmitoylation inhibitor -treated (DIV12-18) WT and *Ppt1^-/-^* neurons and included the results in Figure 11. In line with our previous data, total levels of both GluN2A and PSD-95 in *Ppt1^-/-^* neurons were reduced compared to WT. However, while GluN2B is hyperpalmitoylated in *Ppt1^-/-^*neurons, the basal palmitoylation state of GluN2A and PSD-95 were unchanged between WT and *Ppt1^-/-^* neurons, suggesting that these proteins may not be bona fide PPT1 substrates. Especially, the latter finding is consistent with the previous report showing that PSD-95 is depalmitoylated by ABHD17 but not PPT1 (Yokoi et al., 2016).

As previously reported, we confirmed that WT cells with 2-BP treatment had a reduction in PSD-95 palmitoylation (El-Husseini et al., 2002) but this effect was not observed in *Ppt1^-/-^*neurons. However, the specificity of palmitoylation inhibitors is incompletely understood and compensatory mechanisms may restore PSD-95 palmitoylation due to *chronic low-dose* inhibitor treatment.

Interestingly, chronic palmitoylation inhibitor treatment exhibited unexpected effects on GluN2A palmitoylation state. In WT cells, 2-BP had no effect on GluN2A levels or palmitoylation state (Figure 11K and L), suggesting that the GluN2A may not be the direct substrate of PPT1 in WT neurons. In contrast, 2-BP treatment in *Ppt1^-/-^*neurons robustly increased both the total level and palmitoylation state of GluN2A, resulting in the nearly equal representation of two distinct GluN2A palmitoylated species (Figure 11K and L).

Based on the new results, we have extensively revised the Discussion. In previous versions of our manuscript, we discussed two plausible mechanisms that could explain GluN2B hyperpalmitoylation in *Ppt1^-/-^* neurons. The first possibility is that GluN2B is a PPT1 substrate, and palmitoylation of GluN2B enhances its phosphorylation and consequent surface retention (Hayashi et al., 2009). The second possibility is that the lack of PPT1 function results in hyperpalmitoyaltion of Fyn, which facilitates an overrepresentation of GluN2B at excitatory synapse. In light of the new results on GluN2A, we favor the second possibility, and postulate that 2-BP treatment have suppressed Fyn palmitoylation less than the basal state in *Ppt1^-/-^*neurons and may have accelerated the GluN2B to GluN2A switch. These results are represent a major topic throughout the Discussion section.

2) Changes in synaptic AMPAR currents in PPT1 KO neurons should be reported. Such information is required in order to fully understand the extent to which the observed effects of PPT1 loss are specifically mediated by GluN2B.

Per our previous communication, the editors and reviewers have agreed that this point is beyond the scope of the current manuscript. We therefore aim to report the effects of PPT1 deletion on AMPAR transmission and plasticity in a separate publication. However, we have recognized the reviewers’ comments and have briefly mentioned our relevant preliminary findings pertaining to AMPARs in the Discussion section “Implication for other neurodegenerative diseases”.

3) The claim that the enhancement of slow synaptic NMDR current observed in PPT1 KO neurons is mediated specifically by NR2B-containing NMDARs, should be tested pharmacologically.

We have performed these experiments and have presented them as part of Figure 3. Specifically, we recorded NMDA-EPSCs from WT and *Ppt1^-/-^*visual cortical slices as in Figure 3A-D following bath infusion of the GluN2B-specific antagonist, Ro 25-6981 (3μM). To our surprise, Ro 25-6981 treatment reduced the weighted decay time (τ_w_) of NMDA-EPSCs in both WT and *Ppt1^-/-^*neurons to the same degree (Figure 3E; Results, subsection “NMDAR-mediated EPSCs are altered in *Ppt1^-/-^* visual cortex”). Although this result was not anticipated, plausible explanations for our findings are mentioned in the Results section (see the aforementioned subsection) and discussed further in the context of neurodevelopmental changes in NMDARs (Discussion, subsection “Stagnation of GluN2B to GluN2A subunit switch in the *Ppt1^-/-^*mouse visual cortex”). In brief, we suspect that the synaptic incorporation of GluN1/GluN2A/GluN2B triheteromeric NMDARs is enhanced in *Ppt1^-/-^*neurons in vivo, as these receptors are largely insensitive to Ro 25-6981 treatment but display intermediate decay kinetics. Furthermore, cortical neurons in maturating animals predominantly express NMDARs consisting of two GluN1, one GluN2A and one GluN2B subunits (Sheng et al., 1994; Luo et al., 1997; Tovar and Westbrook, 1999. Thus, these results have improved our working model that PPT1 facilitates GluN2 subunit switch but remain in line with the notion that NMDAR function is dysregulated in *Ppt1^-/-^* visual cortical neurons.

4) The reversal of excitotoxicity in PPT1 KO neurons by palmitoylation inhibitors should be supported by demonstrating also the reversal of altered Ca^2+^ transients under the same treatment conditions.

We have addressed this point in what is now assigned as Figure 10and described the findings in Results (subsection “Palmitoylation inhibitor treatment improves pathological calcium dynamics in *Ppt1^-/-^* neurons”). We treated GCaMP3-transfected *Ppt1^-/-^* neurons with palmitoylation inhibitors 2-BP or cerulenin (1μM, DIV12-18), imaged, and analyzed for calcium dynamics as in Figure 7. We analyzed representative WT and *Ppt1^-/-^,* vehicle-treated cells in parallel to confirm consistency across previous experiments (Video 7) while we used data from Figure 7 and compared them with palmitoylation inhibitor treated cells (Figures 10A-D). Palmitoylation inhibitor treatment partially improved the features of calcium flux in *Ppt1^-/-^*neurons that are accountable for CLN1 pathophysiology. Specifically, the area under the curve (AUC) and diffusion distance of calcium influxes in treated *Ppt1^-/-^*neurons were decreased, although cerulenin treatment did not fully revert these metrics to WT levels. These effects likely contribute to the reduced vulnerability of palmitoylation inhibitor-treated *Ppt1^-/-^*neurons to excitotoxicity (Figure 9F).